# SkipVAR: Accelerating Visual Autoregressive Modeling via Adaptive Frequency-Aware Skipping

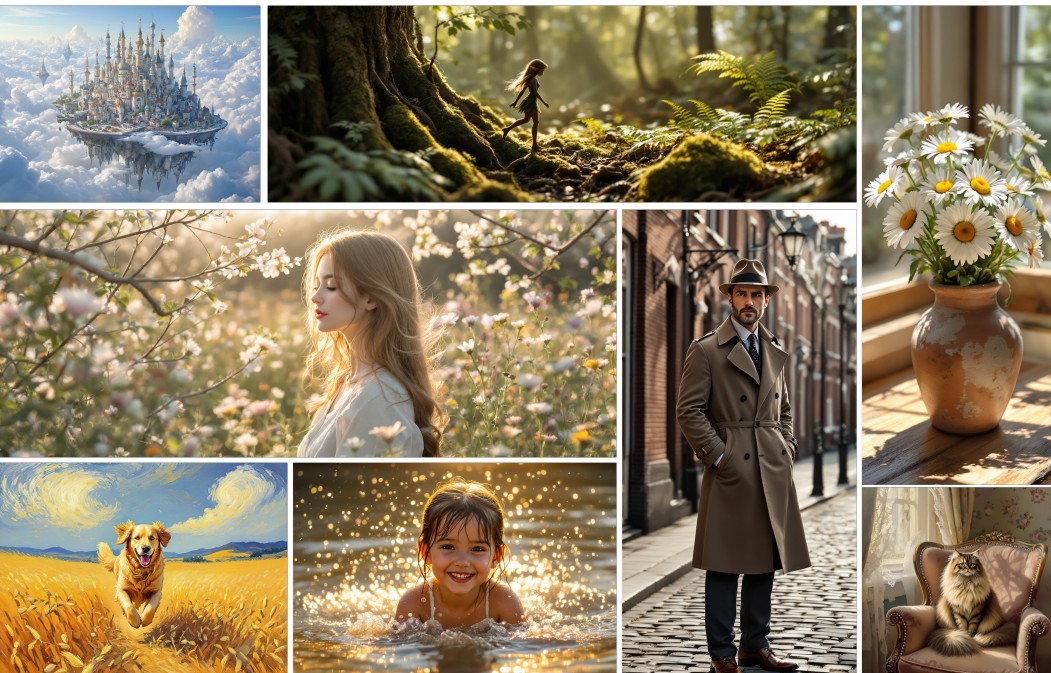

Figure 1: Results generated by Infinity-8B (Han et al., 2024) accelerated with the proposed **SkipVAR**.

## Abstract

Recent studies on Visual Autoregressive (VAR) models have highlighted that high-frequency components, or later steps, in the generation process contribute disproportionately to inference latency. However, the underlying computational redundancy involved in these steps has yet to be thoroughly investigated. In this paper, we conduct an in-depth analysis of the VAR inference process and identify two primary sources of inefficiency: *step redundancy* and *unconditional branch redundancy*. To address step redundancy, we propose an automatic step-skipping strategy that selectively omits unnecessary generation steps to improve efficiency. For unconditional branch redundancy, we observe that the information gap between the conditional and unconditional branches is minimal. Leveraging this insight, we introduce unconditional branch replacement, a technique that bypasses the unconditional branch to reduce computational cost. Notably, we observe that the effectiveness of acceleration strategies varies significantly across different samples. Motivated by this, we propose **SkipVAR**, a sample-adaptive framework that leverages frequency information to dynamically select the most suitable acceleration strategy for each instance. To evaluate the role of high-frequency information, we further introduce multiple high-variation benchmark datasets that evaluate the performance in terms of fine details. Extensive experiments show that SkipVAR achieves over 0.88 average SSIM with up to $1.81\times$ overall acceleration and $2.62\times$ lossless speedup on the GenEval benchmark.

## 1 INTRODUCTION

Visual Autoregressive (VAR) modeling introduces a novel paradigm in image generation by redefining autoregressive learning as a coarse-to-fine "next-scale prediction" process (Tian et al., 2024), diverging from the traditional raster-scan "next-token prediction" approach (Li et al., 2024b; Liu et al., 2024a; Sun et al., 2024; Wang et al., 2024e; Xie et al., 2024a). This methodology enables autoregressive transformers to efficiently learn visual distributions and generalize effectively.

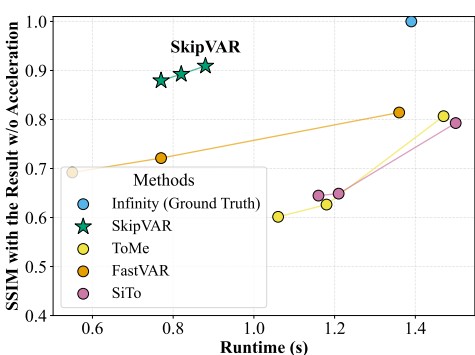

Figure 2: **SSIM v.s. runtime for different acceleration methods. SkipVAR** achieves the best trade-off, maintaining high consistency with faster inference.

Based on the VAR framework, **Infinity** (Han et al., 2024) extends VAR by introducing an infinite-vocabulary tokenizer and bitwise self-correction, enabling high-resolution, photorealistic image generation from text prompts.

Recent advances in accelerating visual autoregressive (VAR) models (Guo et al., 2025; Chen et al., 2024b; Xie et al., 2024b) have revealed that a significant portion of computational overhead arises from the later stages of generation. These findings are consistent with recent analyses indicating that high-frequency components in the generation process contribute disproportionately to inference latency. Although these components often have limited perceptual significance, they require dense computations, thereby becoming a primary bottleneck in inference efficiency. Consequently, identifying and mitigating the computational redundancy in late-stage generation has emerged as a critical direction for enabling scalable and efficient VAR inference.

Nevertheless, both the origins of these inefficiencies and their properties across various samples are still not well understood. In this paper, we conduct an in-depth analysis of the generation dynamics in VAR models and identify two key forms of redundancy that degrade the inference efficiency. The first, *step redundancy*, occurs when later autoregressive steps contribute minimal improvements to the output despite high computational costs, as shown in Figure 3. In such cases, skipping these steps enables significant acceleration without degrading perceptual quality. The second inefficiency, termed *unconditional branch redundancy*, occurs in classifier-free guidance models (Ho & Salimans, 2022) that employ both conditional and unconditional branches, with the latter offering diminishing contributions in later stages. Substituting it with conditional-only generation lowers computational costs without sacrificing output coherence.

Notably, we observe that the effectiveness of acceleration strategies varies significantly across different samples. This variability stems from the fact that the importance of high-frequency information differs substantially from one instance to another. Certain images are highly sensitive to fine-grained structural details and thus demand preservation of high-frequency components for faithful reconstruction. Conversely, other samples can tolerate a reduction in high-frequency computation without noticeable quality degradation, as shown in Figure 4. Despite this heterogeneity, existing VAR acceleration methods (Guo et al., 2025; Chen et al., 2024b; Xie et al., 2024b) use uniform strategies, applying the same scheme to all samples and steps, which ignores individual frequency characteristics, often yielding suboptimal trade-offs between speed and fidelity. These facts suggest the necessity of an adaptive framework that dynamically adjusts the acceleration strategies according to the frequency characteristics of each sample.

To this end, we propose **SkipVAR** in this paper, a sample-adaptive and decision-driven acceleration framework that dynamically selects the most suitable strategy for each instance based on its frequency characteristics. At its core, a lightweight decision model assesses each image's high-frequency sensitivity and selects the optimal VAR-compatible acceleration policy. By exploiting VAR's multi-scale structure and tailoring to content requirements, SkipVAR preserves high-frequency precision where necessary while accelerating tolerant samples, embracing the best of both worlds—efficiency and generation quality—across diverse inputs.

To quantify the role of high-frequency information in VAR models, we design multiple high-variation benchmark datasets that highlight models' sensitivity to fine-grained details. In our experiments, as

shown in Figure 7c, subjective metrics prove unreliable for capturing high-frequency discrepancies. Therefore, we adopt objective similarity metrics to evaluate perceptual fidelity between original and accelerated outputs. Similarity is measured both globally and on high-frequency components to capture fine-detail effects. Based on these considerations, we construct two complementary evaluation sets to systematically assess the impact of high-frequency information: a *frequency-sensitive* set, where quality is highly impacted by high-frequency degradation, and a *frequency-robust* set, where quality remains stable despite reduced high-frequency computation. These benchmarks facilitate clearer evaluation and underscore the need for adaptive methods like SkipVAR.

Through our SkipVAR method, we achieve superior performance in Figure 2, where the curve produced by our approach consistently outperforms a series of token-based methods. Even at a $1.81\times$ acceleration ratio, we maintain high SSIM. Moreover, our approach generalizes seamlessly to the Infinity-8B model, delivering higher acceleration factors. In Figure 1, we present high-quality visual results generated by SkipVAR based on the infinity-8B model, showing the powerful and faster generative ability, including various styles and faces in various image resolutions. Overall, our contributions can be summarized as follows:

- We introduce two efficient acceleration strategies, step-skipping and unconditional branch replacement, and a high-frequency-focused score to emphasize fine-detail preservation.
- We propose SkipVAR, a frequency-sensitive and sample-adaptive framework that adaptively selects an acceleration strategy for each instance.
- To validate the effectiveness of SkipVAR, we perform extensive experiments and user studies to evaluate our approach, which shows our method achieves superior performance compared with existing state-of-the-art VAR acceleration approaches.

## 2 RELATED WORK

**Visual Autoregressive Models.** Visual Autoregressive (VAR) models (Tian et al., 2024; Zhu et al., 2024; Chen et al., 2024a; Ma et al., 2025b; Wang et al., 2024b; Zhang et al., 2024a; Chen et al., 2023; Ma et al., 2022; Feng et al., 2025; Zhang et al., 2025; Ma et al., 2025a; Yan et al., 2025; Wan et al., 2024; Wang et al., 2024c) generate images via coarse-to-fine, multi-scale patch prediction, capturing global layout at early scales and high-frequency detail at finer scales. Infinity (Han et al., 2024) extends this with an infinite-vocabulary tokenizer, bitwise self-correction, and scalable transformer modules, achieving strong GenEval and ImageReward scores while generating 1024×1024 images in 1.4 s. However, token growth at fine scales increases computation and memory, and hierarchical discretization can reduce perceptual fidelity.

**Acceleration in Visual Generation.** Diffusion-based accelerators include distillation (Meng et al., 2023; Ma et al., 2024e;d; Salimans & Ho, 2022), quantization (Li et al., 2023b; Shang et al., 2023), pruning (Bolya et al., 2022; Bolya & Hoffman, 2023; Liu et al., 2025; Ma et al., 2024c; Xiong et al., 2025; Zhu et al., 2025; Fang et al., 2023; Wang et al., 2024a; Zhang et al., 2024b; Zou et al., 2024), and feature caching (Ma et al., 2024b;a; Liu et al., 2024b; Li et al., 2023a). VAR-specific acceleration is less explored: FastVAR (Guo et al., 2025) applies fixed token pruning with cached reuse but ignores per-sample frequency variation. SkipVAR instead makes per-sample, per-scale decisions between step-skipping and conditional branch replacement based on handcrafted frequency-sensitive features, preserving structure and maintaining SSIM above a user-specified threshold. More details and extended discussion are provided in the appendix.

## 3 METHODOLOGY

### 3.1 PRELIMINARY

Autoregressive (AR) models factorize the joint probability of a token sequence $\mathbf{x} = (x_1, \ldots, x_T)$ as $p(\mathbf{x}) = \prod_{t=1}^{T} p(x_t \mid x_{<t})$, and are trained to predict each token from its predecessors. Extending AR to images requires discretizing continuous pixels via a VQ-VAE, which encodes an image $\mathrm{im} \in \mathbb{R}^{H \times W \times 3}$ to $f = E(\mathrm{im})$, then quantizes it to $q = Q(f)$ with a codebook $Z \in \mathbb{R}^{V \times C}$, and reconstructs the image as $\hat{\mathrm{im}} = D(\mathrm{lookup}(Z, q))$, typically under a combination of reconstruction,

perceptual, feature, and adversarial losses. However, the raster-scan autoregressive process breaks spatial locality, incurs $O(n^2)$ decoding steps, and results in $O(n^6)$ operations for an $n \times n$ image. To address this, Visual Autoregressive Modeling (VAR) instead generates coarse-to-fine token maps $\mathbf{R} = (r_1, \ldots, r_K)$, where each $r_k$ is a token grid in $[V]^{h_k \times w_k}$ and the final resolution satisfies $(h_K, w_K) = (h, w)$. The probability is modeled as $p(\mathbf{R}) = \prod_{k=1}^{K} p(r_k \mid r_{<k})$, enabling block-wise causal attention during training and parallel decoding with KV-caching at inference time. This reduces the number of iterations to $O(\log n)$ and computation to $O(n^4)$. Despite these improvements, the rapid increase of fine-scale tokens at high resolutions remains a major computational bottleneck.

## 3.2 MOTIVATION

VAR models (Tian et al., 2024) adopt a coarse-to-fine, multi-scale generation paradigm that improves image quality but increases computational cost at higher scales. This section outlines key motivations for our adaptive approach by examining limitations in current methods.

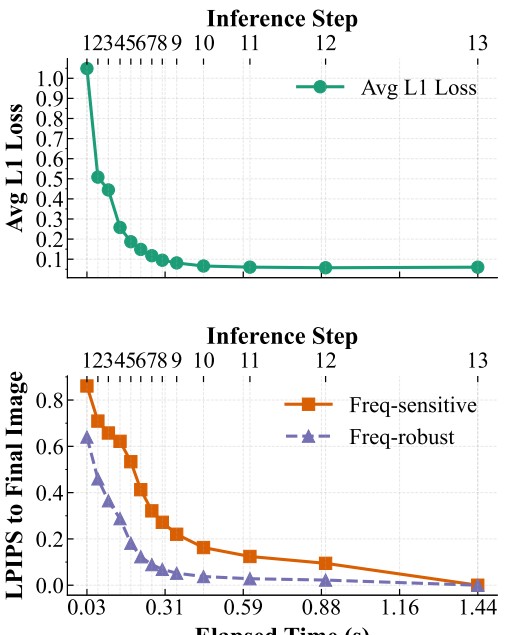

Figure 3: **Top:** L1 loss between conditional and unconditional branches drops over inference steps. **Bottom:** LPIPS to the final image drops, with frequency-robust samples consistently lower than frequency-sensitive ones.

**1. Redundancy in High-Frequency Generation.** Early scales capture global structure, while later scales refine high-frequency details (Tian et al., 2024; Guo et al., 2025). However, such refinement is costly and yields diminishing perceptual returns. Figure 3 shows that although later inference steps dominate runtime, LPIPS improvements plateau and image changes become visually minor. Both conditional and unconditional losses decrease steadily, but separately computing these branches imposes significant overhead without commensurate quality gains.

**2. Variability in High-Frequency Sensitivity.** Images differ in their dependence on high-frequency detail. For instance, an anime-style headshot exhibits minimal changes across steps, while a realistic portrait requires ongoing refinement (Figure 4). Existing VAR acceleration methods (Guo et al., 2025; Chen et al., 2024b; Xie et al., 2024b) apply uniform strategies, risking wasted computation or quality loss. Figure 3 shows that frequency-robust samples have consistently lower LPIPS than frequency-sensitive ones, highlighting the need for sample-specific acceleration.

**3. Limitations of Existing Benchmarks.** Common metrics like ImageReward (Xu et al., 2023) and CLIP Score (Hessel et al., 2021) emphasize overall plausibility but neglect fine-detail fidelity crucial in later VAR stages. As shown in Figure 7c, these subjective metrics often fail to capture detail degradation. Consequently, blurred or less detailed outputs may still score highly, masking acceleration effects. We advocate comparing accelerated outputs to full-step references using objective metrics SSIM and LPIPS (Zhang et al., 2018), supplemented by high-frequency variants (`SSIM-HF`, `LPIPS-HF`) to better evaluate fine-detail preservation. Moreover, we construct frequency-sensitive and frequency-robust datasets to make such differences more evident.

In sum, redundancy in high-frequency generation, variability in sample sensitivity, and benchmark limitations motivate an adaptive, image-specific acceleration strategy. Our proposed SkipVAR addresses these challenges by dynamically tailoring generation per image, enhancing computational efficiency without compromising perceptual quality.

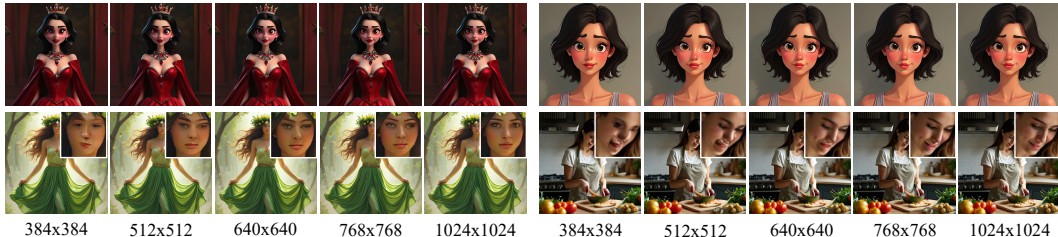

| 384x384 | 512x512 | 640x640 | 768x768 | 1024x1024 | 384x384 | 512x512 | 640x640 | 768x768 | 1024x1024 |

Figure 4: **Visualization of Post-Scaling Steps for Samples with Varying Sensitivity.**

### 3.3 METHOD

We propose **SkipVAR**, a post-training adaptive acceleration framework. SkipVAR exploits the inherent redundancy in multi-scale generation by dynamically selecting acceleration strategies on a per-sample basis, guided by the significance of high-frequency components. Our method aims to substantially reduce inference cost without compromising perceptual quality.

**Overview of SkipVAR.** Given a multi-scale VAR model generating an image over $K$ refinement steps, we designate a step $N$ (typically early, e.g., $N = 10$) as the **decision step**. After $N$, acceleration strategies may be selectively applied to the remaining steps $N + 1, N + 2, \ldots, K$. To guide decisions, we use decoded images from step $N$ and the cached result from step $N-1$. These outputs are downsampled and analyzed in image space to estimate high-frequency importance, with downsampled decoding to improve efficiency. The extracted features are then fed into a lightweight decision model $\mathcal{D}$, which outputs the optimal acceleration policy for the remaining steps on a per-sample basis.

**Acceleration Strategies.** SkipVAR incorporates two practical strategies to exploit redundancy:

- **Skip Strategy:** For the Skip Strategy, we target frequency-robust samples by selecting a specific acceleration step beyond which decoding stops early. As illustrated in Figure 5, we directly decode the output at that step without proceeding to further resolution refinements.
- **Uncond-Branch-Replace Strategy:** For the unconditional branch replacement Strategy, we focus on frequency-sensitive samples. As shown in Figure 7b, we leverage the convergence between the conditional and unconditional branches in the later stages. By reusing the conditional output in place of the unconditional one, we effectively cut computation in half. Compared to the Skip Strategy, the additional cost of computing the conditional branch ensures that frequency-sensitive samples still receive sufficient high-frequency information.

**High-Frequency Importance Estimation.** To characterize a sample's sensitivity to high-frequency information, we compute two complementary indicators:

HIGH FREQUENCY DIFFERENCE (HF_DIFF). To quantify local variations in high-frequency structures during the generation process, we apply a $3 \times 3$ Sobel operator $\mathcal{S}$ to the grayscale images decoded at two consecutive steps, $N$ and $N-1$. The high-frequency difference is then defined as the $\ell_1$ distance between their corresponding Sobel responses:

$$\text{HF\_Diff} = \|\mathcal{S}(I_N) - \mathcal{S}(I_{N-1})\|_1,\tag{1}$$

where $I_N$ and $I_{N-1}$ denote the decoded grayscale images at steps $N$ and $N-1$, respectively. A smaller value of HF_Diff indicates that the high-frequency content has stabilized between the two steps, implying diminishing returns from further refinement. Therefore, when HF_Diff falls below a threshold, the sample is considered less sensitive to high-frequency variations, making it a strong candidate for early termination or aggressive acceleration.

HIGH FREQUENCY RATIO (HF_RATIO). To complement the local difference measurement, we further assess the global frequency composition of the decoded image using the Fourier transform. Specifically, we define the high-frequency ratio as:

$$\text{HF\_Ratio} = \frac{\sum_{(u,v) \in \mathcal{H}} |\mathcal{F}(I_N)|_{uv}}{\sum_{(u,v)} |\mathcal{F}(I_N)|_{uv} + \epsilon},\tag{2}$$

where $\mathcal{F}(I_N)$ denotes the magnitude of the shifted Fourier transform of the grayscale image $I_N$, and $\mathcal{H}$ is the set of high-frequency components beyond a radius $\rho$ (typically $\rho = 0.4$) from the spectrum

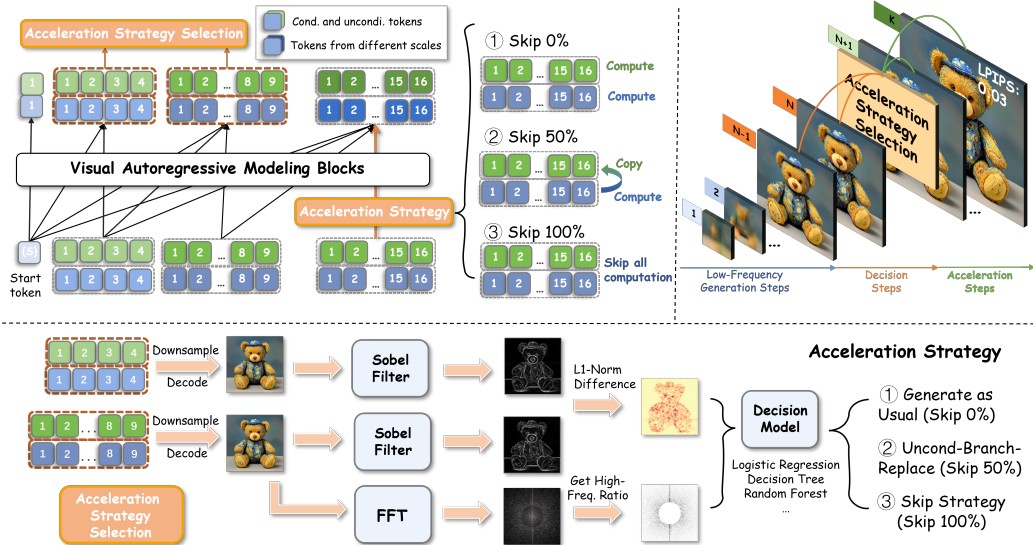

Figure 5: **Top Left:** Overall framework. At the Decision Step, an acceleration strategy is selected and applied to subsequent steps. For clarity, we omit the propagation during the acceleration steps. **Top Right:** Visualization of SkipVAR's pipeline. We divide the generation process into *Low-Frequency Generation Steps*, *Decision Steps*, and *Acceleration Steps* to emphasize the roles of each stage. **Bottom:** Detailed framework for acceleration strategy selection. After downsampling and decoding the results into image space, we compute HF_Diff and HF_Ratio. These indicators are then fed into decision models to determine the optimal acceleration strategy.

center. A small constant $\epsilon$ avoids division by zero. A lower HF_Ratio indicates the image has less high-frequency content, making it more resilient to early truncation or simplified refinement, and thus a strong candidate for aggressive acceleration. By combining HF_Diff and HF_Ratio, frequency-sensitive and frequency-robust samples are roughly linearly separable, allowing simple machine learning models to effectively distinguish them, as illustrated in Figure 7b.

**Decision Models.** To achieve low-latency inference decisions, we employ simple, efficient classifiers for the decision model $\mathcal{D}$. Specifically, we consider: 1. **Decision Tree (DT):** A tree-based model that partitions the feature space with learned thresholds, offering fast and interpretable decisions. 2. **Random Forest (RF):** An ensemble of decision trees that aggregates predictions across multiple randomized subsets to improve robustness and generalization. 3. **Logistic Regression (LR):** A linear model that estimates the

Table 1: **Evaluation on different datasets.** Gray: DrawBench, Orange: Paintings, Cyan: Photo.

| Method | Spd. | MACs | IRwd | Clip |
|---|---|---|---|---|
| Infinity | 1.00× | 31097 | 0.8618 | 0.2702 |
| +SkipVAR@0.88 | 1.58× | 18134 | 0.8688 | 0.2701 |
| +SkipVAR@0.86 | 1.70× | 16467 | 0.8570 | 0.2705 |
| +SkipVAR@0.84 | 1.81× | 15237 | 0.8575 | 0.2704 |
| Infinity | 1.00× | – | 1.2767 | 0.2746 |
| +SkipVAR@0.84 | 1.53× | – | 1.2742 | 0.2745 |
| Infinity | 1.00× | – | 0.9991 | 0.2618 |
| +SkipVAR@0.88 | 1.55× | – | 0.9876 | 0.2621 |
| +SkipVAR@0.84 | 1.75× | – | 0.9854 | 0.2625 |

probability of applying acceleration strategies via a sigmoid activation over the feature vector. Based on the experiments in Appendix E, we ultimately adopt Logistic Regression for all subsequent experiments due to its efficiency and interpretability. Labels are assigned according to the most aggressive acceleration strategy that preserves perceptual quality above a predefined SSIM threshold, ensuring each sample is paired with its maximum safe acceleration. Feature-label pairs are standardized, and classifiers are trained using a standard 80/20 split. As shown in Table 5 and Appendix G, decision models trained on a single class maintain strong performance on other datasets, demonstrating that our frequency-aware mechanism captures patterns that generalize beyond the training domain. The effectiveness of SkipVAR arises from its strategic design rather than the complexity of the model.

## 4 EXPERIMENTS

### 4.1 EXPERIMENTS SETUP

**Setup and Evaluation.** We evaluate SkipVAR on the open-source Infinity-2B VAR model, generating images up to 1024×1024. The acceleration decision is made at step 10 based on two features:

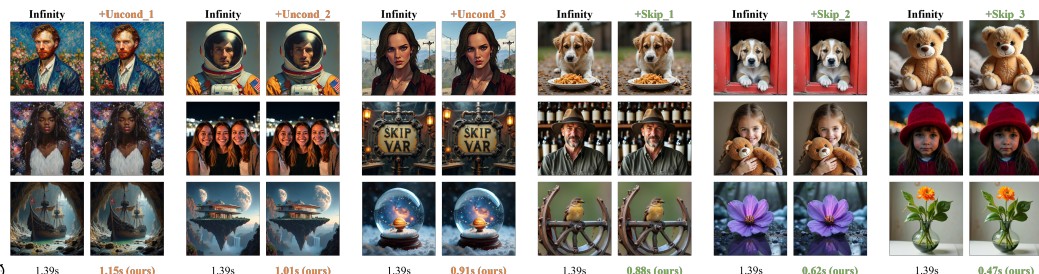

Figure 6: **Image generations under different acceleration strategies.** Strategies are selected by our decision model. Simpler images use faster strategies with less high-frequency detail; complex ones use slower strategies to preserve more detail. Images are grouped and sorted by strategy speed.

HF_Diff (via third-order Sobel) and HF_Ratio ($\rho = 0.4$). A decision model trained on the *People* category of MJHQ30K (Li et al., 2024a) selects the most aggressive fidelity-preserving strategy subject to SSIM thresholds $\{0.88, 0.86, 0.84\}$. This design demonstrates that strong performance can be achieved even when the decision model is trained on a single class, highlighting that our frequency-aware mechanism generalizes effectively to other domains. During evaluation, acceleration is applied to autoregressive steps only (11–13), and generation quality is assessed via ImageReward (Xu et al., 2023), GenEval (Ghosh et al., 2023), HPSv2 (Wu et al., 2023), GPT Scores (Achiam et al., 2023), SSIM (Wang et al., 2004), LPIPS (Zhang et al., 2018), and FID (Heusel et al., 2017), while efficiency is measured by Acceleration Ratio, Inference Time, and FLOPS. Experiments are conducted on a single NVIDIA RTX 4090D GPU. Additional details are provided in Appendix D.

## 4.2 MAIN RESULTS

**Comparison on subjective scores.** 1. **ImageReward and CLIP Score Evaluation:** We evaluate SkipVAR on DrawBench (200 prompts) and on the Paintings and Photo subsets of HPSV2 (Wu et al., 2023) (800 images each) using ImageReward (Xu et al., 2023) and CLIP Score (Hessel et al., 2021). As shown in Table 1, SkipVAR@0.84, SkipVAR@0.86, and SkipVAR@0.88 all achieve near-identical scores to the original Infinity model across all three datasets, demonstrating that our method maintains high subjective quality even on stylistically diverse image sets. 2. **Quality–Speed Tradeoff on HPSV2:** On the full HPSV2 benchmark (Table 2a), SkipVAR@0.84 attains a $1.73\times$ speedup for only a 0.08-point drop in average score (from 30.49 to 30.41), preserving Anime, Concept-art, Paintings,

Table 2: **Comparison on multiple benchmarks.** (a) Evaluation on the HPSv2 benchmark. (b) GPT-based evaluation on frequency-sensitive and frequency-robust datasets; blue rows correspond to **Frequency-sensitive** dataset, green rows correspond to **Frequency-robust** dataset.

(a) HPSv2 evaluation.

| Methods | Speedup | Ani. | Art | Paint | Photo |
|---|---|---|---|---|---|
| Infinity | $1.00\times$ | 31.70 | 30.45 | 30.40 | 29.43 |
| +SkipVAR@0.84 | $1.73\times$ | 31.59 | 30.27 | 30.49 | 29.30 |

(b) GPT-based frequency evaluation.

| Methods | Lat. | Spd. | Aes | Align | SSIM |
|---|---|---|---|---|---|
| Infinity | 1.39 | – | 88.36 | 86.69 | – |
| +SkipVAR@0.84 | 1.09 | $1.28\times$ | 88.11 | 87.59 | 0.849 |
| Infinity | 1.39 | – | 87.18 | 89.75 | – |
| +SkipVAR@0.84 | 0.70 | $1.99\times$ | 87.02 | 90.98 | 0.905 |

and Photo quality within 0.2 points. Figure 6 visualizes representative outputs ordered by acceleration level, showing that simpler inputs trigger aggressive skipping while complex scenes retain unconditional branch replacement in line with human perceptual expectations. 3. **GenEval Benchmark Performance:** On the GenEval benchmark (Ghosh et al., 2023) (Table 3), SkipVAR@0.86 delivers a $1.77\times$ speedup with only a $\sim$1% drop in overall score (from 0.71 to 0.70). Furthermore, our SkipVAR-hybrid (w/o DM) configuration—applying skip to steps 12–13 and unconditional branch replacement to steps 10–11—achieves a $2.62\times$ speedup while matching or exceeding baseline two-objective, position, and color attribute scores, underscoring the robustness of our frequency-aware acceleration design. 4. **Adaptation to Frequency Sensitivity:** We categorize HPSV2 images into *frequency-sensitive* and *frequency-robust* sets based on SSIM sensitivity at late inference steps. Table 2b shows that SkipVAR@0.84 applies the slower unconditional branch replacement strategy for sensitive samples—preserving SSIM at 0.8493—and the faster skip strategy for robust samples—achieving SSIM of 0.9051—yielding up to a $1.99\times$ speedup without perceptible subjective degradation. Additionally, we observe that the GPT score does not decrease due to acceleration and

Table 3: **Comparison of different generation models on the GenEval (Ghosh et al., 2023) benchmark.** ● Diffusion models, ■ Autoregressive models, ★ Infinity model. Prompt rewriting was applied to the Infinity model in our experiments.

| Methods | Params | Speedup | Two Obj. | Position | Color Attri. | Overall |
|---|---|---|---|---|---|---|
| ●SDXL | 2.6B | - | 0.74 | 0.15 | 0.23 | 0.55 |
| ●PixArt-Sigma | 0.6B | - | 0.62 | 0.14 | 0.27 | 0.55 |
| ●SD3-medium | 2.0B | - | 0.74 | 0.34 | 0.36 | 0.62 |
| ■LlamaGen | 0.8B | - | 0.34 | 0.07 | 0.04 | 0.32 |
| ■Show-o | 1.3B | - | 0.80 | 0.31 | 0.50 | 0.68 |
| ★Infinity | 2.0B | - | 0.85 | 0.39 | 0.55 | 0.71 |
| ★+FastVAR | 2.0B | 2.53× | 1.00 | 0.81 | 0.55 | 0.6828 |
| ★+ToMe | 2.0B | 1.18× | 0.94 | 0.68 | 0.28 | 0.5364 |
| ★+SiTo | 2.0B | 1.15× | 0.98 | 0.76 | 0.42 | 0.62228 |
| ★+SkipVAR@0.88 | 2.0B | 1.50× | 0.83 | 0.39 | 0.59 | 0.71 |
| ★+SkipVAR@0.86 | 2.0B | 1.77× | 0.84 | 0.36 | 0.58 | 0.70 |
| ★+SkipVAR-hybrid (w/o DM) | 2.0B | 2.62× | 0.84 | 0.39 | 0.60 | 0.72 |

even shows a slight increase in the Align metric, indicating that our approach maintains high semantic consistency while achieving significant speedup.

**Comparison on objective metrics.** Because subjective scores often fail to capture high-frequency detail differences, we evaluate our method using rigorous objective metrics as follows (see Appendix O for further discussion): 1. **SSIM and LPIPS:** We assess acceleration effects with SSIM and LPIPS under a fixed random seed for direct comparability to the Infinity baseline. On DrawBench (Table 4), SkipVAR maintains SSIM above each model's training threshold and outperforms token-based approaches such as ToMe (Bolya & Hoffman, 2023), SiTo (Zhang et al., 2024b), and FastVAR (Guo et al., 2025). For instance, ToMe@0.05—applying 5% token merging at the final step—yields SSIM < 0.81, whereas SkipVAR's holistic acceleration better preserves global fidelity. 2. **SSIM-HF Analysis:** To capture fine-detail fidelity, we introduce `SSIM-HF`, computed on high-frequency regions only. Also on DrawBench (Table 4), ToMe@0.05 shows overall SSIM just 8% lower than SkipVAR@0.84, but `SSIM-HF` degrades by 34%, demonstrating the severe impact of suboptimal token selection on fine textures. 3. **MJHQ30K FID Performance:** On the MJHQ30K benchmark (Table 5), SkipVAR@0.84 achieves a $1.88\times$ speedup with low FID. Even on the "Art" subset—where high-frequency detail is paramount—it delivers a $1.58\times$ acceleration without significant quality loss.

## 4.3 EMPIRICAL STUDIES

Table 4: **Quantitative comparison on the Draw-Bench dataset. ToMe** and **SiTo** use the default ratio of {0.5,0.5,0.5}, while **FastVAR** is reproduced with official settings. Notations such as **+ToMe@0.05** and **+SiTo@0.05** indicate that a 0.05 ratio is applied solely at the final step.

| Methods | Spd. | SSIM | SSIM-HF | LPIPS | LPIPS-HF |
|---|---|---|---|---|---|
| Infinity | 1.00× | - | - | - | - |
| +SkipVAR@0.88 | 1.58× | 0.909 | 0.343 | 0.049 | 0.182 |
| +SkipVAR@0.86 | 1.70× | 0.892 | 0.310 | 0.058 | 0.193 |
| +SkipVAR@0.84 | 1.81× | 0.879 | 0.288 | 0.065 | 0.200 |
| +HFDiffOnly | 1.72× | 0.875 | 0.282 | 0.067 | 0.202 |
| +HFRatioOnly | 1.82× | 0.865 | 0.267 | 0.071 | 0.207 |
| +ToMe@0.05 | 0.95× | 0.807 | 0.191 | 0.104 | 0.238 |
| +ToMe | 1.18× | 0.626 | 0.111 | 0.352 | 0.413 |
| +SiTo@0.05 | 0.93× | 0.793 | 0.182 | 0.109 | 0.241 |
| +SiTo | 1.15× | 0.649 | 0.117 | 0.287 | 0.367 |
| +FastVAR@0.05 | 1.02× | 0.814 | 0.196 | 0.098 | 0.233 |
| +FastVAR | 2.53× | 0.692 | 0.127 | 0.204 | 0.309 |
| +SkipVAR-hybrid | 2.62× | 0.792 | 0.183 | 0.108 | 0.239 |

**Feature Design and Objective Analysis.** Robust decision-making requires features that capture fine-detail sensitivity while being insensitive to luminance and contrast variations. Sole reliance on Sobel-based $\text{HF\_Diff}$, which measures local high-frequency stability across refinement steps, can misclassify dark but smooth images as high-frequency. Conversely, using $\text{HF\_Ratio}$ alone tends to adopt more aggressive acceleration strategies for samples whose $\text{HF\_Ratio}$ is moderate yet still frequency-sensitive, resulting in lower similarity. Figure 7a further confirms that the intermediate $\text{HF\_Ratio}$ reliably estimates final high-frequency content, justifying its use at the decision step. To address these issues, we combine $\text{HF\_Diff}$ and $\text{HF\_Ratio}$ into a two-dimensional feature vector capturing both local edge stability and global texture richness. As Table 4 shows, $\text{HF\_Ratio}$ alone slightly improves speedup but worsens generation quality, while $\text{HF\_Diff}$ alone offers more stable perceptual results. Combining both yields the best trade-off across speed, SSIM/LPIPS, including on high-frequency regions. This validates that integrating both features balances their biases and enables more reliable acceleration decisions across diverse scenes.

**Evaluation of SkipVAR Decision Models and Generalization.** We evaluate SkipVAR across different classifiers, datasets, and model scales to assess its robustness. Three lightweight decision

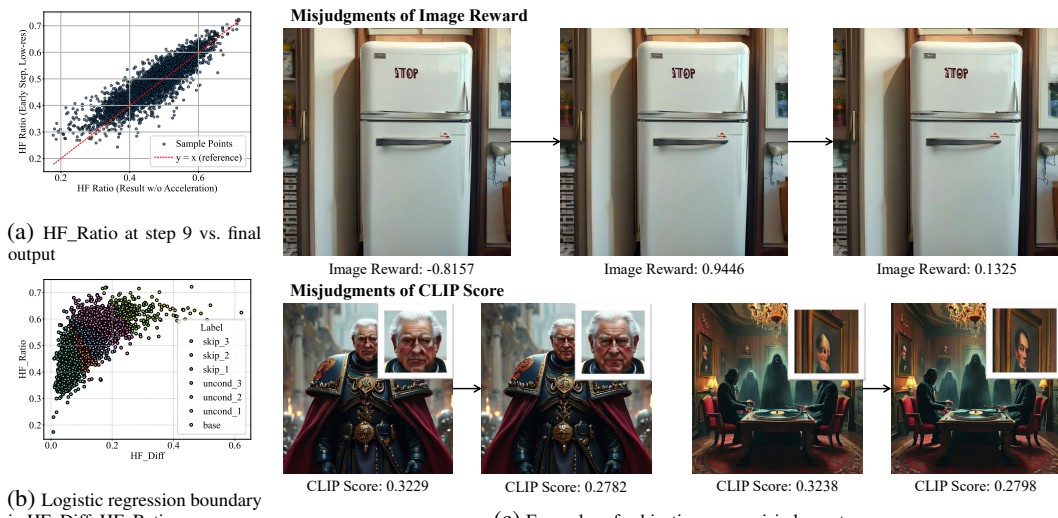

(a) HF_Ratio at step 9 vs. final output

(b) Logistic regression boundary in HF_Diff–HF_Ratio space

(c) Examples of subjective score misjudgments

Figure 7: **(a)** Scatter of points lie along $y = x$, showing reliable low-res Fourier estimates. **(b)** Decision labels: `skip_3` = skip last 3 steps, `uncond_2` = replace uncond branch for last 2 steps. **(c)** Example images: HF-Robust (top) and HF-Sensitive (bottom).

models—Logistic Regression (LR), Random Forest (RF), and Decision Tree (DT)—perform comparably in terms of visual quality, with LR slightly more efficient and interpretable (see Appendix E). Cross-category evaluation demonstrates that classifiers trained on either People or Animals datasets perform similarly on the DrawBench benchmark (Appendix E), confirming robust decisions across domains. Moreover, mixed-domain training further improves generalizability, enabling more confident acceleration choices. Importantly, SkipVAR decision models trained on the 2B Infinity model transfer effectively to the larger 8B model, yield-

Table 5: **Generation quality on MJHQ30K**

| Dataset | Speedup | FID | SSIM | LPIPS |
|---|---|---|---|---|
| Food | 1.88× | 3.56 | 0.8675 | 0.0780 |
| Fashion | 1.76× | 3.64 | 0.8713 | 0.0765 |
| Plants | 1.60× | 2.69 | 0.8690 | 0.0649 |
| Art | 1.58× | 3.02 | 0.8663 | 0.0704 |

ing substantial acceleration while preserving image quality (Appendix F). Finally, as shown in Table 5, SkipVAR achieves up to $1.88\times$ speedup on datasets like *Food*, with SSIM above 0.86. Evaluation on additional MJHQ30K categories (Landscape, Indoor, Logo, Vehicles) further confirms strong generalization across domains with different frequency characteristics (Appendix G). These results collectively validate that SkipVAR decisions are robust across classifiers, datasets, and model scales.

**Decision Point Selection and Sensitivity Analysis.** Profiling reveals that the last few decoding steps (11–13) dominate runtime, despite most token distributions having stabilized. We fix the *decision point* at step 10 ($N = 10$) and only accelerate the final three steps, balancing computational efficiency and structural completeness. Steps 11–13 account for 69% of total inference time, and LPIPS improvements taper off beyond step 10, even for frequency-sensitive cases (Figure 3). Sensitivity analysis shows that choosing step 9 slightly increases speedup at the cost of quality, while step 11 improves similarity but reduces speedup. Step 10 empirically strikes the best trade-off across datasets (see Appendix I for detailed results).

## 5 CONCLUSION

In this paper, we present **SkipVAR**, a training-free, sample-adaptive acceleration framework for Visual Autoregressive (VAR) models. We observe that there are existing step redundancy and unconditional branch redundancy in VAR. To address these issues, we propose an automatic step-skipping strategy for late steps and an approach that replaces the unconditional branch to reduce unnecessary computation. Importantly, we find that the effectiveness of these acceleration strategies varies across samples. Motivated by this, SkipVAR dynamically leverages frequency information to select the most suitable strategy for each instance. To assess high-frequency sensitivity, we introduce dedicated benchmark datasets, highlighting the impact of fine details. Extensive results demonstrate the effectiveness of frequency-aware training-free adaptive acceleration for efficient generation.

## ETHICS STATEMENT

This work adheres to the ICLR Code of Ethics. Our research does not involve human subjects, personal or sensitive data, or any proprietary datasets. All datasets used are publicly available and cited appropriately. We do not anticipate any direct societal risks, such as privacy violations or security concerns, arising from our proposed method. The contributions focus on algorithmic improvements for accelerating visual autoregressive models, which we believe pose minimal potential for harmful misuse. No conflicts of interest or external sponsorship have influenced this work.

## REPRODUCIBILITY STATEMENT

We have taken concrete steps to ensure that our results are fully reproducible. The complete source code and scripts have been included in the supplementary materials in an anonymous form. The full training procedure for the decision models is detailed in Appendix C, while the experimental setup, including hyperparameters and evaluation settings, is provided in Appendix D. All datasets used in this study are standard benchmarks and have been cited accordingly in the main text. The evaluation metrics employed are explicitly described in their respective sections. Together, these materials and descriptions ensure that independent researchers can reproduce our findings without ambiguity.

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

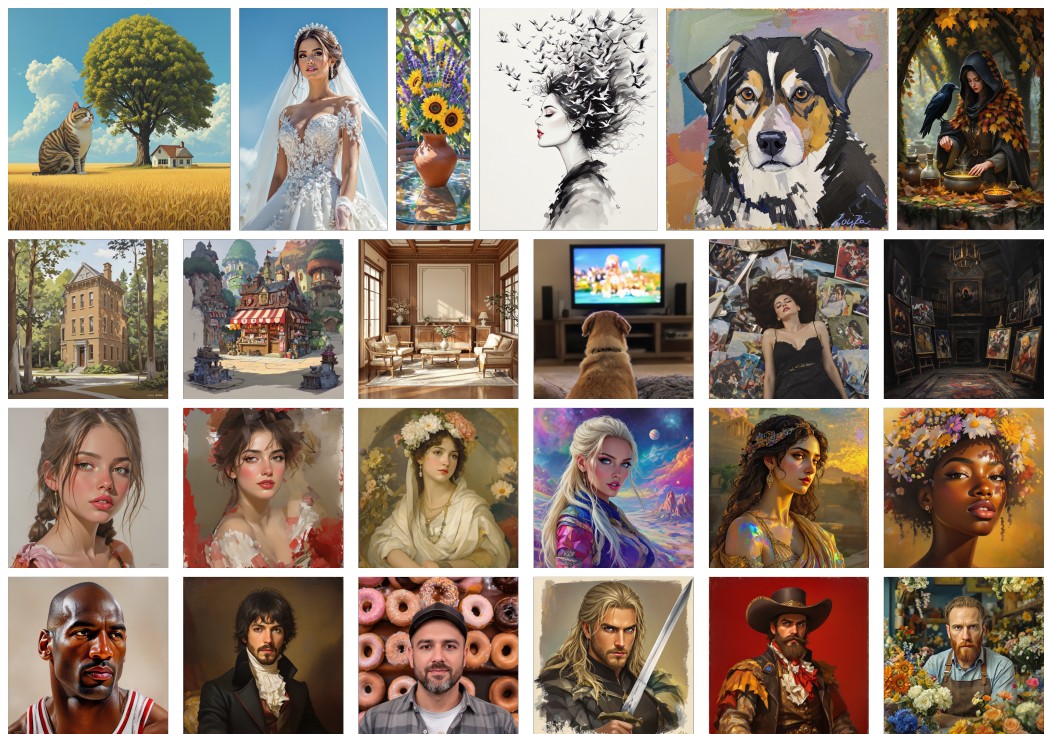

Figure 8: More results obtained by acceleration with the SkipVAR method.

## A  THE USE OF LARGE LANGUAGE MODELS (LLMS)

In accordance with the ICLR 2026 policy on the use of large language models (LLMs) in paper writing, we disclose that LLMs were employed solely for language-related purposes. Specifically, we used LLMs to assist with grammar correction, sentence polishing, and improving readability of the text. Importantly, LLMs were not used to generate novel research ideas, design experiments, conduct analysis, or draw conclusions. All substantive contributions, including methodology, experiments, and analysis, were conducted entirely by the authors.

## B  RELATED WORK DETAILS

**Visual Autoregressive Models.**  Visual Autoregressive (VAR) models (Tian et al., 2024) depart from token-by-token generation by employing a coarse-to-fine, multi-scale next-patch strategy and diffusion transformer (Zhu et al., 2024; Chen et al., 2024a; Ma et al., 2025b; Wang et al., 2024b; Zhang et al., 2024a; Chen et al., 2023; Ma et al., 2022; Feng et al., 2025; Zhang et al., 2025; Ma et al., 2025a; Yan et al., 2025; Wan et al., 2024; Wang et al., 2024c). Early scales capture global layout and structure, while finer scales synthesize high-frequency textures. Infinity (Han et al., 2024) extends VAR with three innovations: an infinite-vocabulary tokenizer and classifier for bitwise token prediction, bitwise self-correction for detail refinement, and theoretically unbounded transformer scaling. These yield state-of-the-art GenEval and ImageReward scores (0.96 vs. 0.87) while generating 1024×1024 images in 1.4 s. However, token counts grow exponentially at fine scales, causing substantial computation and memory overhead, and hierarchical discretization can accumulate errors that degrade perceptual fidelity.

**Acceleration Techniques in Visual Generation.**  Diffusion model acceleration includes distillation (Meng et al., 2023; Ma et al., 2024e;d; Salimans & Ho, 2022), quantization (Li et al., 2023b; Shang et al., 2023), pruning (Bolya et al., 2022; Bolya & Hoffman, 2023; Liu et al., 2025; Ma et al., 2024c; Xiong et al., 2025; Zhu et al., 2025; Fang et al., 2023; Wang et al., 2024a; Zhang et al., 2024b; Zou et al., 2024), and feature caching (Ma et al., 2024b;a; Liu et al., 2024b; Li et al., 2023a).

These do not directly extend to hierarchical, patch-based VAR generation. FastVAR (Guo et al., 2025) accelerates VAR via post-training token pruning with cached reuse, but applies a fixed ratio across images and scales, ignoring per-sample variation in high-frequency importance and conditional information, which can cause wasted computation or perceptual degradation.

SkipVAR departs from uniform acceleration by making per-sample, per-scale decisions between aggressive step-skipping and conservative unconditional branch replacement based on handcrafted frequency-sensitivity features. This mechanism preserves structural coherence, avoids pruning artifacts, and maintains SSIM above a user-defined threshold, enabling efficient inference across diverse inputs.

## C    DETAILED TRAINING AND LABELING PROCEDURE

**Dataset and Features.**    The decision model is trained using the **People** class of the MJHQ30K dataset (Li et al., 2024a). While the full dataset contains multiple categories, we deliberately use only the People class to demonstrate that strong performance can be achieved without multi-class data, highlighting the method's generalization ability. For each sample, two handcrafted features are extracted: HF_Diff, capturing local Sobel edge stability, and HF_Ratio, describing global frequency content in the Fourier domain.

**Label Assignment.**    Labels are assigned at a designated decision step ($N$=10 by default, though other steps are supported). For each sample, we generate:

1. a base version with standard decoding,
2. skip variants, where steps after $N$ are skipped (e.g., `skip_2`, `skip_3`),
3. unconditional-to-conditional variants, replacing unconditional branches after $N$ with conditional ones (e.g., `uncond_2`, `uncond_3`).

Each accelerated version is compared with the base using SSIM. The label is the most aggressive strategy that satisfies a predefined quality threshold: if a skip variant succeeds, the label is `skip_x`; if skipping fails but unconditional replacement succeeds, the label is `uncond_x`; otherwise, the label is `base`. This ensures each sample receives the maximum safe acceleration without perceptible quality loss.

**Model Training.**    The resulting feature-label pairs are standardized, and classifiers are trained with an 80/20 train/validation split. Multiple models were tested; all main experiments use Logistic Regression, emphasizing that the effectiveness of SkipVAR stems from its strategic design rather than model complexity.

## D    SETUP AND EVALUATION DETAILS

**Model and Acceleration Mechanism.**    We evaluate our proposed acceleration strategy on the open-source Infinity-2B model, a state-of-the-art Bitwise Visual AutoRegressive (VAR) model capable of generating high-resolution images up to 1024×1024 pixels. The acceleration decision in SkipVAR is made at the 10th inference step, based on two handcrafted frequency-aware features: the high-frequency difference (computed using a third-order Sobel operator) and the high-frequency ratio (thresholded with $\rho = 0.4$). A decision model is trained on the *people* category of the MJHQ30K dataset (Li et al., 2024a), which consists of 3,000 diverse high-resolution images including both stylized and photo-realistic human figures. This model selects the most aggressive yet fidelity-preserving acceleration strategy—either skipping steps or replacing unconditional branches—subject to SSIM thresholds $\{0.88, 0.86, 0.84\}$.

**Evaluation Protocol and Metrics.**    During evaluation, acceleration is applied only to autoregressive decoding steps (11–13), excluding VAE decoding time for fair comparisons. Generation quality is measured using ImageReward (Xu et al., 2023), GenEval (Ghosh et al., 2023), HPSv2 (Wu et al., 2023), GPT Scores (Achiam et al., 2023), SSIM (Wang et al., 2004), LPIPS (Zhang et al., 2018), and FID (Heusel et al., 2017). Efficiency metrics include Acceleration Ratio, Inference Time, and

FLOPS. All experiments are conducted on a single NVIDIA RTX 4090D GPU with 24GB VRAM, using default settings for consistency.

# E    ABLATION AND CROSS-DOMAIN RESULTS

We compare three lightweight classifiers for SkipVAR: Logistic Regression (LR), Random Forest (RF), and Decision Tree (DT). All models maintain favorable speed-quality trade-offs. LR is slightly more efficient and interpretable. Cross-category evaluation shows that training on either People or Animals datasets yields robust performance on DrawBench.

Table 6: Comparison of decision models and cross-domain generalization.

| Methods | Speedup | SSIM | SSIM-HF | LPIPS | LPIPS-HF |
|---|---|---|---|---|---|
| Infinity | 1.00× | - | - | - | - |
| +SkipVAR@0.84_lr | 1.81× | 0.8793 | 0.2881 | 0.0646 | 0.1997 |
| +SkipVAR@0.84_rf | 1.73× | 0.8833 | 0.2938 | 0.0623 | 0.1977 |
| +SkipVAR@0.84_dt | 1.73× | 0.8832 | 0.2954 | 0.0623 | 0.1975 |
| +SkipVAR@0.84_lr_animals | 1.79× | 0.8806 | 0.2892 | 0.0641 | 0.1992 |
| +SkipVAR@0.84_rf_animals | 1.74× | 0.8853 | 0.2985 | 0.0616 | 0.1964 |
| +SkipVAR@0.84_dt_animals | 1.74× | 0.8875 | 0.3022 | 0.0603 | 0.1953 |

# F    CROSS-MODEL TRANSFER: 2B → 8B

Applying SkipVAR models trained on 2B Infinity to the 8B model demonstrates effective cross-model transfer. Despite differences in model scale, acceleration and visual quality remain high. Since the original 4090 GPU could not run the 8B model, we utilized VGPU-48GB to obtain the reported results in the table.

Table 7: Quantitative Results on DrawBench using SkipVAR decision models trained on 2B, applied to Infinity-8B.

| Method | Latency | Speedup | SSIM | LPIPS | SSIM-HF | LPIPS-HF |
|---|---|---|---|---|---|---|
| Infinity_8B | 3.38 s | — | — | — | — | — |
| +SkipVAR@0.88_via_2B | 1.86 s | 1.81× | 0.901 | 0.0406 | 0.3885 | 0.1765 |
| +SkipVAR@0.86_via_2B | 1.74 s | 1.94× | 0.8795 | 0.0476 | 0.3472 | 0.1857 |
| +SkipVAR@0.84_via_2B | 1.64 s | 2.06× | 0.8592 | 0.0549 | 0.3121 | 0.1955 |

# G    EVALUATION ON DIVERSE MJHQ30K CATEGORIES

We evaluate SkipVAR on four categories with varying frequency characteristics. Despite training on a single People dataset, SkipVAR maintains competitive FID, high SSIM, and low LPIPS, confirming strong cross-domain generalization.

Table 8: SkipVAR evaluation on diverse MJHQ30K categories.

| Dataset | Method | FID | SSIM | LPIPS | SSIM-LF | SSIM-HF | LPIPS-LF | LPIPS-HF |
|---|---|---|---|---|---|---|---|---|
| Landscape | +SkipVAR@0.84 | 2.2316 | 0.8516 | 0.0717 | 0.9291 | 0.3130 | 0.0624 | 0.1980 |
| Indoor | +SkipVAR@0.84 | 2.6772 | 0.8709 | 0.0659 | 0.9116 | 0.3500 | 0.0706 | 0.1965 |
| Logo | +SkipVAR@0.84 | 2.7801 | 0.9078 | 0.0555 | 0.9225 | 0.3614 | 0.0653 | 0.1850 |
| Vehicles | +SkipVAR@0.84 | 2.4594 | 0.8711 | 0.0663 | 0.9125 | 0.3243 | 0.0689 | 0.1972 |

# H    RESULTS WITH MIXED TRAINING DATA

In addition to the results reported in the main text, we further investigate the impact of training the decision classifier on a more diverse dataset. Specifically, we employ a mixed dataset for training, aiming to refine the decision boundaries and improve robustness across different domains.

The classifier remains lightweight—logistic regression—with inference time under 0.4 milliseconds per sample, thus introducing negligible computational overhead. While training on the *People* class alone already yields strong generalization, incorporating the other class provides additional benefits, particularly for domains where the single-class model tended to be conservative. For example, on Indoor scenes, SkipVAR achieves **1.67**× speedup with SSIM 0.8540 and LPIPS 0.0745.

Table 9: Results of SkipVAR with classifier trained on a mixed dataset.

| Dataset | Method | Speedup | FID | SSIM | LPIPS | SSIM-HF | LPIPS-HF |
|---------|--------|---------|-----|------|-------|---------|----------|
| Landscape | +SkipVAR@0.84_appendOtherDataset | 1.61× | 2.5246 | 0.8371 | 0.0797 | 0.2927 | 0.2044 |
| Indoor | +SkipVAR@0.84_appendOtherDataset | 1.67× | 3.0108 | 0.8540 | 0.0745 | 0.3250 | 0.2053 |
| Vehicles | +SkipVAR@0.84_appendOtherDataset | 1.66× | 2.8251 | 0.8545 | 0.0753 | 0.2977 | 0.2065 |

These findings suggest that incorporating multiple classes into the training data yields more balanced decision boundaries and further enhances the robustness of SkipVAR across diverse domains.

## I    SENSITIVITY ANALYSIS OF DECISION STEP

We conducted a sensitivity analysis to evaluate the effect of varying the decision point between steps 9, 10, and 11. As shown in Table 10, choosing an earlier step (e.g., step 9) slightly increases acceleration but marginally lowers image quality, while a later step (e.g., step 11) improves SSIM and LPIPS but reduces speedup. Step 10 represents a "knee point" that balances speed and perceptual quality across datasets.

Table 10: Performance of SkipVAR across different decision steps.

| Dataset | Decision Step | Speedup | SSIM | LPIPS |
|---------|---------------|---------|------|-------|
| DrawBench | Step 9 | 1.84x | 0.8767 | 0.0675 |
| DrawBench | Step 10 | 1.81x | 0.8793 | 0.0646 |
| DrawBench | Step 11 | 1.59x | 0.8939 | 0.0555 |
| Photo | Step 9 | 1.80x | 0.8792 | 0.0703 |
| Photo | Step 10 | 1.75x | 0.8814 | 0.0678 |
| Photo | Step 11 | 1.65x | 0.8971 | 0.0576 |
| Paintings | Step 9 | 1.53x | 0.8662 | 0.0625 |
| Paintings | Step 10 | 1.53x | 0.8660 | 0.0620 |
| Paintings | Step 11 | 1.44x | 0.8825 | 0.0533 |

These results confirm that step 10 is an effective decision point across datasets: it avoids redundant high-resolution decoding while maintaining structural and perceptual quality, providing a practical trade-off for SkipVAR acceleration.

## J    ANALYSIS OF HANDCRAFTED FEATURES

To ensure low inference overhead and practical efficiency, SkipVAR adopts two handcrafted features, `HF_Diff` and `HF_Ratio`, chosen for their simplicity, interpretability, and minimal computational cost. These features are frequency-aware yet lightweight, making them well suited for real-time acceleration scenarios.

We further analyze the robustness of our approach under different Fourier mask radius settings $\rho$. As shown in Table 11, when $\rho$ ranges from 0.1 to 0.4, both quality metrics and speedup remain stable, confirming that our method is not sensitive to the exact choice of $\rho$. At smaller $\rho$ values, frequency-sensitive and frequency-robust images exhibit similar high-frequency ratios, leading to more conservative acceleration. At larger $\rho$ values, this distinction becomes clearer, enabling more aggressive acceleration for frequency-robust images.

This analysis shows that handcrafted features, despite their simplicity, provide stable and effective decision signals. By avoiding reliance on learned representations, SkipVAR achieves strong generalization while introducing almost no computational overhead.

Table 11: Performance of SkipVAR under different $\rho$ settings.

| Methods | Speedup | SSIM | SSIM-HF | LPIPS | LPIPS-HF |
|---------|---------|------|---------|-------|----------|
| Infinity | $1.00\times$ | - | - | - | - |
| +SkipVAR@0.84, $\rho = 0.1$ | $1.75\times$ | 0.8833 | 0.2916 | 0.0624 | 0.1980 |
| +SkipVAR@0.84, $\rho = 0.25$ | $1.77\times$ | 0.8804 | 0.2882 | 0.0638 | 0.1994 |
| +SkipVAR@0.84, $\rho = 0.4$ | $1.81\times$ | 0.8793 | 0.2881 | 0.0646 | 0.1997 |

## K  SCALABILITY OF ACCELERATION STRATEGIES.

Although our current acceleration strategy mainly relies on step-skipping or unconditional branch replacement, more acceleration schemes can be integrated in principle. Both strategies already employ decision models to determine how many steps to accelerate, rather than applying acceleration uniformly across all steps. By leveraging the decision model to distinguish between high-sensitivity and low-sensitivity images, we can assign different acceleration levels accordingly—applying more aggressive acceleration to low-sensitivity images, while adopting more conservative strategies for high-sensitivity ones. This flexibility enables finer-grained, image-aware control over inference efficiency, potentially unlocking greater acceleration gains while preserving output quality.

Motivated by this, we further explored *hybrid acceleration strategies* within the decision space. For example, applying unconditional branch replacement in the first steps and skipping in later steps better matches sample complexity and leads to faster generation. To this end, we extended our strategy space to include hybrid decoding paths alongside pure skipping paths, and trained the decision model to select adaptively among them. As shown in Table 12, the resulting variant (**SkipVAR@0.84-hybrid**) achieves slightly improved speedup (1.85× vs. 1.81×) while maintaining comparable SSIM and LPIPS values, demonstrating that the decision model can naturally incorporate hybrid variants where appropriate without sacrificing perceptual quality.

Table 12: Impact of Decision Space Hybridization on SkipVAR Performance.

| Methods | Speedup | SSIM | SSIM-HF | LPIPS | LPIPS-HF |
|---------|---------|------|---------|-------|----------|
| Infinity | $1.00\times$ | - | - | - | - |
| +SkipVAR@0.84 | $1.81\times$ | 0.8793 | 0.2881 | 0.0646 | 0.1997 |
| +SkipVAR@0.84-hybrid (w DM) | $1.85\times$ | 0.8774 | 0.2878 | 0.0667 | 0.2011 |

In summary, the decision model remains a crucial and effective component of SkipVAR, enabling balanced acceleration with high-quality output, while also allowing integration of hybrid strategies that further enhance scalability.

## L  LIMITATIONS OF SSIM-BASED EVALUATION.

In our experiments, we train decision models using SSIM-based thresholds to guide acceleration strategies. This objective metric helps align accelerated outputs with those generated by the original model. However, we acknowledge a key limitation: a lower SSIM score does not necessarily indicate perceptible quality degradation to the human eye. For instance, in face close-ups where high-frequency features dominate, even minor acceleration can significantly lower SSIM while leaving visual perception largely unaffected.

We further investigated alternative perceptual metrics following reviewer feedback. Specifically, we examined ImageReward, but found that it struggles to capture fine-detail quality. In some cases, it cannot distinguish between images with the same semantic content but different levels of detail fidelity. For example, in our supplementary material, nearly indistinguishable images to humans receive widely varying scores (e.g., from $-0.82$ to $0.94$ to $0.13$).

We also experimented with LPIPS-based thresholds as a replacement for SSIM. While LPIPS better reflects human perception in certain scenarios, the resulting acceleration ratios were lower, and overall perceptual similarity of the generated images was still inferior compared to SSIM-based thresholds.

Table 13: Performance of SkipVAR Variants with Different Training Thresholds.

| Methods | Speedup | SSIM | SSIM-HF | LPIPS | LPIPS-HF |
|---|---|---|---|---|---|
| Infinity | 1.00× | - | - | - | - |
| +SkipVAR@0.88 | 1.58× | 0.9092 | 0.3429 | 0.0488 | 0.1823 |
| +SkipVAR@0.86 | 1.70× | 0.8924 | 0.3100 | 0.0577 | 0.1925 |
| +SkipVAR@0.84 | 1.81× | 0.8793 | 0.2881 | 0.0646 | 0.1997 |
| +SkipVAR@LPIPS=0.05 | 1.36× | 0.9296 | 0.3886 | 0.0375 | 0.1680 |
| +SkipVAR@LPIPS=0.07 | 1.52× | 0.9017 | 0.3212 | 0.0507 | 0.1863 |
| +SkipVAR@LPIPS=0.09 | 1.78× | 0.8712 | 0.2698 | 0.0661 | 0.2029 |

Table 13 summarizes the performance of SkipVAR under different training thresholds, showing that SSIM-based criteria provide a better balance between perceptual fidelity and acceleration.

In summary, while SSIM is not a perfect proxy for human visual judgment, our experiments show that it remains a more stable and effective training signal compared to LPIPS and ImageReward. Future work could explore combining SSIM with perceptual metrics to achieve more human-aligned decision boundaries, particularly for high-frequency sensitive cases.

## M    SKIPVAR IS FUNDAMENTALLY SAMPLE-ADAPTIVE — NOT A UNIFORM TRADE-OFF

Unlike fixed or step-reduction baselines, **SkipVAR dynamically adapts the acceleration strategy per sample**, guided by its frequency characteristics. This ensures that complex or detail-sensitive inputs receive more conservative treatment, while simpler samples are accelerated more aggressively.

To further illustrate this property, we report additional results on the frequency-sensitive (`HF_high_similarity`) and frequency-robust (`HF_low_similarity`) subsets used in Table 2b. The results are summarized in Table 14, showing how **one-size-fits-all acceleration methods** (e.g., SiTo, ToMe, FastVAR) suffer from substantial quality drops when applied uniformly across diverse samples. In contrast, **SkipVAR@0.84**, despite its higher average speedup, consistently maintains significantly better SSIM and LPIPS scores—particularly on challenging frequency-sensitive subsets.

Table 14: Comparison on frequency-sensitive (`HF_high_similarity`) and frequency-robust (`HF_low_similarity`) subsets. SkipVAR achieves superior quality–speed trade-offs through per-sample adaptivity.

| Dataset | Method | Speedup | SSIM | LPIPS |
|---|---|---|---|---|
| Frequency-sensitive | Infinity | — | — | — |
| | +SkipVAR@0.84 | 1.28× | 0.8493 | 0.0617 |
| | +SkipVAR-hybrid (w/oDM) | 2.62× | 0.5605 | 0.1952 |
| | +SiTo | 1.16× | 0.3811 | 0.4169 |
| | +ToMe | 1.21× | 0.3794 | 0.4657 |
| | +FastVAR | 2.52× | 0.4433 | 0.3316 |
| Frequency-robust | Infinity | — | — | — |
| | +SkipVAR@0.84 | 1.99× | 0.9051 | 0.0689 |
| | +SkipVAR-hybrid (w/oDM) | 2.62× | 0.8922 | 0.0772 |
| | +SiTo | 1.16× | 0.7721 | 0.2455 |
| | +ToMe | 1.21× | 0.7388 | 0.3155 |
| | +FastVAR | 2.53× | 0.8116 | 0.1628 |

From Table 14, we observe that:

- On Frequency-sensitive samples, **SiTo, ToMe, and FastVAR all result in SSIM $< 0.45$**, indicating severe perceptual degradation. In contrast, **SkipVAR@0.84 preserves a strong SSIM of 0.8493 and low LPIPS of 0.0617**, even with a higher speedup than the others;

- On Frequency-robust samples, SkipVAR further improves both fidelity and speedup, demonstrating **smooth quality adaptation** across frequency regimes.

For completeness, we also report the variant **SkipVAR-hybrid (w/oDM)**, which removes the decision model while retaining SkipVAR's two acceleration strategies (step skipping and unconditional branch replacement). Even in this setting, the method outperforms other fixed baselines under similar speedups.

These results underscore that **SkipVAR's per-sample adaptivity, coupled with its core acceleration strategies, enables superior quality–speed trade-offs**, particularly under challenging frequency-sensitive scenarios.

Table 15: Comparison of Super-Resolution (SR) and direct skipping for late steps in VAR model

| Method | SSIM ↑ | LPIPS ↓ | SSIM-HF ↑ | LPIPS-HF ↓ |
|---|---|---|---|---|
| SR-13 | 0.8412 | 0.1405 | 0.2128 | 0.3055 |
| Skip-13 | 0.8621 | 0.0632 | 0.2596 | 0.2011 |
| SR-12-13 | 0.8078 | 0.1576 | 0.1841 | 0.3189 |
| Skip-12-13 | 0.8218 | 0.0875 | 0.2099 | 0.2233 |
| SR-11-13 | 0.7712 | 0.1797 | 0.1599 | 0.3347 |
| Skip-11-13 | 0.7802 | 0.1172 | 0.1726 | 0.2467 |

## N    EVALUATING STEP REPLACEMENT VIA SUPER-RESOLUTION IN VAR MODELS

To investigate whether super-resolution (SR) models can compensate for the removal of later steps in the VAR model (specifically steps 11, 12, and 13), we conducted experiments comparing direct skipping and SR-based recovery methods. We adopt FreqFormer (Wang et al., 2024d) as the SR backbone in our evaluation. As shown in Table 15, the use of super-resolution models (denoted with the "_2x" suffix) results in consistently lower performance across multiple metrics such as SSIM, Cosine similarity, and LPIPS, when compared to their non-SR counterparts. Despite reducing inference time, the super-resolution models fail to effectively replicate the generation quality of the omitted steps. This suggests that directly replacing the later autoregressive steps with SR recovery cannot maintain the original model's fidelity and thus does not bring meaningful improvements.

## O    MISJUDGMENTS OF COMMON SUBJECTIVE METRICS ON IMAGES

We observe that widely used subjective evaluation metrics—namely ImageReward, CLIP Score, and GPT Score—often misjudge the importance of high-frequency details. When evaluating images whose overall structure has been largely completed, these metrics fail to capture subtle but perceptually significant fine textures.

Specifically:

- For *HF-Robust* examples, human observers perceive negligible differences, yet ImageReward exhibit large fluctuations (see Figure 7c).

- For *HF-Sensitive* examples, the removal of high-frequency details visibly degrades the image, but paradoxically the restoration of these details sometimes leads to a *decrease* in the subjective scores.

- GPT Score exhibits significant instability: even when asked to distinguish between two images with clearly differing high-frequency details—among a pool of 300 comparison images—GPT consistently fails to identify which image contains superior high-frequency information.

These observations highlight the limitations of current subjective metrics in reflecting the contribution of fine visual details. Figure 7c provides representative examples.

## P    MULTI-METRIC EVALUATION OF SKIPVAR

This section provides extended quantitative results supporting the comparative evaluation between SkipVAR and existing acceleration methods, including FastVAR. To avoid reliance on any single metric, we report a diverse set of semantic, perceptual, and distribution-level measurements. All experiments are conducted on the MJHQ30K benchmark and DrawBench prompts as described in the main paper.

### P.1    FID ACROSS CATEGORIES

Table 16 summarizes the FID evaluation across the *Food*, *Art*, and a balanced *All* subset (3,000 prompts; 300 per category). SkipVAR consistently achieves lower FID than FastVAR at comparable or slightly higher acceleration ratios, indicating improved perceptual distribution quality independent of the SSIM-based threshold used in SkipVAR's decision model.

Table 16: FID comparison across categories. Lower is better.

| Methods | Category | Speedup | FID |
|---|---|---|---|
| FastVAR_0.5_0.6_0.8 | Food | 1.53× | 6.84 |
| SkipVAR@0.84 | Food | 1.88× | **3.56** |
| FastVAR | Food | 2.53× | 6.38 |
| SkipVAR-hybrid (w/oDM) | Food | 2.62× | **4.35** |
| FastVAR_0.5_0.6_0.8 | Art | 1.50× | 7.62 |
| SkipVAR@0.84 | Art | 1.58× | **3.02** |
| FastVAR | Art | 2.53× | 8.65 |
| SkipVAR-hybrid (w/oDM) | Art | 2.62× | **5.16** |
| FastVAR_0.5_0.6_0.8 | All | 1.50× | 7.08 |
| SkipVAR@0.84 | All | 1.72× | **3.13** |
| FastVAR | All | 2.53× | 7.04 |
| SkipVAR-hybrid (w/oDM) | All | 2.62× | **4.82** |

### P.2    SEMANTIC ALIGNMENT VIA CLIP SCORE

Table 17 reports CLIPScore results on DrawBench prompts. SkipVAR maintains semantic consistency with the input descriptions, performing comparably to FastVAR across all acceleration settings. These results confirm that SkipVAR's skipping strategy does not impair semantic alignment.

Table 17: CLIPScore on DrawBench. Higher is better.

| Methods | Speedup | CLIPScore |
|---|---|---|
| Infinity (baseline) | – | 0.2712 |
| SkipVAR@0.84 | 1.84× | 0.2711 |
| FastVAR | 2.53× | 0.2715 |
| SkipVAR-hybrid (w/oDM) | 2.62× | 0.2714 |

### P.3    PERCEPTUAL QUALITY VIA HPSV2

Table 18 presents HPSv2 perceptual quality measurements across four visual domains. SkipVAR achieves a stronger speed–quality balance than FastVAR, particularly in frequency-sensitive domains (e.g., Anime, Paintings), where FastVAR tends to degrade fine-grained details.

Table 18: HPSv2 perceptual quality comparison. Higher is better.

| Methods | Speedup | Anime | Concept-Art | Paintings | Photo | Avg |
|---------|---------|-------|-------------|-----------|-------|-----|
| Infinity | 1.00× | 31.70 | 30.45 | 30.40 | 29.43 | 30.49 |
| SkipVAR@0.84 | 1.73× | 31.59 | 30.27 | 30.49 | 29.30 | 30.41 |
| FastVAR | 2.80× | 31.06 | 29.87 | 29.98 | 28.85 | 29.94 |
| SkipVAR-hybrid (w/oDM) | 2.98× | 31.49 | 30.22 | 30.34 | 29.33 | 30.34 |

## P.4 SUMMARY

Across FID, CLIPScore, HPSv2, and other perceptual metrics not shown here for brevity, the following observations hold:

- SkipVAR improves multiple quality indicators beyond SSIM, including distribution-level (FID) and perceptual (HPSv2) metrics.
- SkipVAR provides a more favorable speed–quality trade-off compared to FastVAR.
- The hybrid variant further enhances performance, achieving both higher acceleration and competitive or improved perceptual quality.

These multi-metric results provide comprehensive evidence that SkipVAR delivers robust acceleration while preserving or improving image quality across diverse visual domains and evaluation metrics.

## Q ON THE NECESSITY OF ADAPTIVE DECISION MODELING

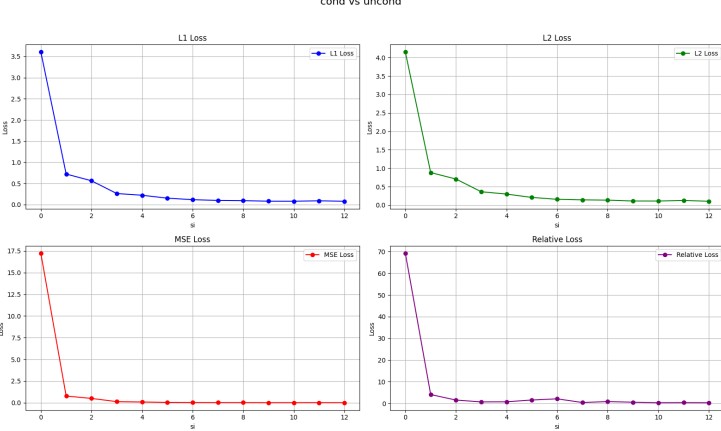

Figure 9: Convergence of conditional and unconditional branches in later steps, measured via L1, L2, MSE and relative losses.

This section provides additional evidence supporting the role of the adaptive decision model in SkipVAR. The adaptive mechanism is required because samples differ substantially in their frequency sensitivity: while frequency-robust prompts can tolerate aggressive skipping, frequency-sensitive prompts exhibit significant degradation when a unified acceleration schedule is applied.

The decision model is introduced to distinguish between frequency-sensitive and frequency-robust samples and to assign appropriate skipping behaviors. This separation is crucial for mixed real-world datasets, where prompts naturally span a spectrum of frequency characteristics. The practical impact of this adaptive identification is illustrated through a comparison with a simple heuristic rule and a random policy on DrawBench:

Although the heuristic achieves comparable SSIM, it consistently produces lower acceleration because it cannot reliably identify robust samples. Conversely, the random baseline demonstrates that incorrect identification—even at similar nominal speed—results in substantial quality degradation.

Table 19: Comparison of adaptive vs. heuristic vs. random selection on DrawBench.

| Methods | Speedup | SSIM | LPIPS |
|---|---|---|---|
| SkipVAR@0.86 | **1.70×** | **0.8924** | 0.057 |
| Simple Logic heuristic | 1.66× | 0.8931 | 0.057 |
| Random selection | 1.70× | 0.8451 | 0.057 |

These results indicate that neither conservative fixed schedules nor simple rules can replicate the selective, sample-aware behavior of the adaptive model. The decision mechanism is therefore essential for achieving safe acceleration on sensitive inputs while maximizing speed on robust ones, enabling SkipVAR to operate effectively across diverse prompt types.

## R    ROLE OF UNCONDITIONAL BRANCH AND SKIP POLICY

SkipVAR employs a dual-branch design, where the unconditional branch primarily guides global structure during early steps, and later steps focus on high-frequency refinement. In these later steps, both conditional and unconditional losses steadily decrease, indicating convergence of the two branches. Visualizations of L2, MSE, and relative losses (Figure 9) further quantify this convergence.

To evaluate the necessity of unconditional branch replacement, we ablated a skip-only variant in which the decision model is constrained to predict only skip actions. Results on standard benchmarks are summarized in Table 20.

Table 20: Skip-only variant without unconditional branch replacement.

| Methods | Speedup | SSIM | LPIPS | ImageReward | ClipScore |
|---|---|---|---|---|---|
| Infinity | - | - | - | 0.8881 | 0.2708 |
| SkipVAR_skip-only | 1.72× | 0.9064 | 0.0555 | 0.8826 | 0.2721 |

These results demonstrate that the learned skipping policy alone achieves significant acceleration with negligible quality loss. While unconditional branch replacement remains safe due to branch convergence in late steps, it is not strictly required. This validates SkipVAR's design: most speedup arises from identifying and skipping redundant high-frequency computations, while the dual-branch structure ensures robust guidance when needed.

