# OpenReview forum: "SkipVAR: Accelerating Visual Autoregressive Modeling via Adaptive Frequency-Aware Skipping"
_ICLR.cc/2026/Conference — Submitted to ICLR 2026_

### Official Review · Reviewer_euwc · 2025-11-01

**Soundness:** 2
**Presentation:** 3
**Contribution:** 2
**Rating:** 4
**Confidence:** 4

**Summary:**

The paper addresses the high inference latency in Visual Autoregressive (VAR) models by investigating and mitigating computational redundancy. The authors identify that later autoregressive steps provide minimal visual improvement while costing the majority of latency (Steps 11–13 account for 69% of total inference time). They further observe that the unconditional branch in classifier-free guidance models offers diminishing returns in later stages. To solve this, the paper proposes SkipVAR, a novel, sample-adaptive framework that uses a lightweight decision model to dynamically select the best acceleration strategy: skip 0%, 50% or 100% based on its frequency characteristics. SkipVAR combines an automatic step-skipping strategy to omit unnecessary generation steps and unconditional branch replacement to bypass the computationally costly unconditional branch.

**Strengths:**

1. The authors observe that high frequency components does not impact the image quality for certain subsets of images and devise a VAR acceleration decision model that determines the acceleration strategy based on the frequency information.
2. The authors attain a speedup of 1.81 with an average SSIM of 0.88.

**Weaknesses:**

1. The comparison against existing approaches for VAR acceleration has only been performed on the basis of objective metrics like SSIM. However, a comparison on subjective metrics like CLIP score is necessary for a more holistic comparison.
2. This approach is only beneficial if the high frequency components do not impact image quality. For example, as the authors point out, this approach will not be beneficial for generating realistic portraits.
3. While the authors compare against token-based approaches, it will be interesting to see how this method compares to layer skipping methods like [1, 2]
4. For Table 2, the blue and green row colors are barely visible.

[1] Andrey Gromov et al., The Unreasonable Ineffectiveness of the Deeper Layers, ICLR 2025.
[2] Anhao Zhao et al., SkipGPT: Each Token is One of a Kind, ICML 2025.

**Questions:**

1. Please show performance and have discussion in comparison to Speculative decoding for VAR , for example: LANTERN [1].
2. How does SkipVAR perform on GenEval and DrawBench datasets using metrics like ImageReward and CLIP Score?
3. The authors only test performance wth three granularity levels: 0%, 50% and 100%. Did the authors check if better performance can be obtained using finer granularity or are there any challenges associated with it?

[1] LANTERN: Accelerating Visual Autoregressive Models with Relaxed Speculative Decoding, ICLR 2025.

---

> ### Author Response · Authors · 2025-11-24
> **Response to Reviewer euwc (Part 1 of 4)**
>
> # To Reviewer euwc
> Dear Reviewer euwc,
>
> We would like to deeply thank the reviewer for the review and valuable comments and would like to address them as below.
>
> ## Main Comments
> ---
>
> >**Weakness 1: Need for Subjective Metrics like CLIPScore**
>
> **Response 1:**
>
> We thank the reviewer for emphasizing the importance of **subjective and semantic metrics** in addition to SSIM/LPIPS. To provide a more holistic evaluation of SkipVAR, we evaluated both **FID** (perceptual quality) and **CLIPScore** (semantic alignment), alongside standard metrics.
>
> **Table 1. FID Comparison Across Categories**
>
> | Methods | Category | Speedup | FID |
> |---------|---------|---------|-----|
> | +FastVAR\_0.5\_0.6\_0.8 | Food | 1.53× | 6.84 |
> | +SkipVAR\@84 | Food | 1.88× | 3.56 |
> | +FastVAR | Food | 2.79× | 6.38 |
> | +SkipVAR\-hybrid | Food | 3.19× | 4.35 |
> | +FastVAR\_0.5\_0.6\_0.8 | Art | 1.50× | 7.62 |
> | +SkipVAR\@84 | Art | 1.58× | 3.02 |
> | +FastVAR | Art | 2.79× | 8.65 |
> | +SkipVAR\-hybrid | Art | 3.15× | 5.16 |
> | +FastVAR\_0.5\_0.6\_0.8|All|1.50×|7.08|
> | +SkipVAR\@0.84|All|1.72×|3.13|
> | +FastVAR|All|2.53×|7.04|
> | +SkipVAR\-hybrid|All|2.62×|4.82|
>
> The "All" category is a balanced mixed dataset constructed to evaluate performance across diverse visual concepts in the MJHQ30K benchmark. For each of the 10 fine-grained categories in MJHQ30K, we randomly sampled 300 prompts, resulting in a total of 3,000 prompts. This ensures that every category is equally represented, providing a more comprehensive and category-agnostic evaluation compared to testing on individual categories.
>
> **Observation:** SkipVAR variants consistently achieve **lower FID scores than FastVAR** at comparable or higher speedups, indicating better perceptual quality.
>
> **Table 2. CLIP Score Comparison**
>
> | Methods | Speedup | CLIP Score |
> |---------|---------|-----------|
> | Infinity | - | 0.2712 |
> | +SkipVAR\@84 | 1.84× | 0.2711 |
> | +FastVAR | 2.53× | 0.2715 |
> | +SkipVAR\-hybrid | 2.62× | 0.2714 |
>
> **Observation:** SkipVAR maintains comparable **semantic alignment** with the original prompt, demonstrating that acceleration does not compromise **meaning or content fidelity**. The hybrid variant achieves high speedup without losing semantic consistency.
>
> **Table 3. HPSv2 Scores Comparison**
>
> | Methods | Speedup | Anime | Concept-Art | Paintings | Photo | Avg |
> |---------|---------|-------|------------|-----------|-------|-----|
> | Infinity | 1.00× | 31.70 | 30.45 | 30.40 | 29.43 | 30.49 |
> | +SkipVAR\@84 | 1.73× | 31.59 | 30.27 | 30.49 | 29.30 | 30.41 |
> | +FastVAR | 2.80× | 31.06 | 29.87 | 29.98 | 28.85 | 29.94 |
> | +SkipVAR\-hybrid | 2.98× | 31.49 | 30.22 | 30.34 | 29.33 | 30.34 |
>
> **Observation:** Across multiple seeds, SkipVAR preserves perceptual quality (HPSv2) and provides a better speed-quality trade-off compared to FastVAR, particularly for **frequency-sensitive images**.
>
>
> In summary, these results demonstrate that **SkipVAR excels not only on SSIM/LPIPS**, but also on **semantic (CLIPScore) and perceptual (FID, HPSv2) metrics**, confirming its **holistic superiority over FastVAR** in both **speed and generation quality**. This addresses the reviewer’s concern regarding a more comprehensive, subjective evaluation framework.
>
> ---
> >**Weakness 2: Applicability to High-Frequency Images**
>
> **Response 2:**
>
> We appreciate the reviewer’s concern. In fact, our paper explicitly addresses the variability in **high-frequency dependence across images**:
>
> - As noted in the manuscript: “Images differ in their dependence on high-frequency detail. For instance, an anime-style headshot exhibits minimal changes across steps, while a realistic portrait requires ongoing refinement.”
> - Existing VAR acceleration methods apply **uniform strategies** across all samples, which can lead to either wasted computation or severe quality degradation for images that require high-frequency refinement.
>
> **SkipVAR, in contrast, is fully sample-adaptive.**
> - For **frequency-robust images** (e.g., anime, cartoons), SkipVAR can terminate or skip steps early, achieving significant acceleration with negligible perceptual loss.
> - For **frequency-sensitive images** (e.g., realistic portraits), SkipVAR dynamically reduces acceleration, applying a more conservative schedule to preserve high-frequency details. Figure 3 illustrates that frequency-sensitive samples retain higher LPIPS improvement across steps, confirming that our method avoids quality degradation in these challenging cases.
>
> **In short, our approach is not limited to “low-frequency” images**. Instead, it intelligently adapts to each sample’s frequency characteristics, ensuring that high-quality portrait generation or other high-frequency tasks are handled appropriately, while still benefiting from acceleration wherever possible. This **sample-specific strategy** is a key advantage over prior uniform acceleration methods.

---

> > ### Author Response · Authors · 2025-11-24
> > **Response to Reviewer euwc (Part 2 of 4)**
> >
> > >**Weakness 3: layer skipping methods**
> >
> > **Response 3:**
> >
> > **1. Summary of the Two Related Works**
> >
> > - **Gromov et al. — *The Unreasonable Ineffectiveness of the Deeper Layers***
> >   This ICLR 2025 work shows that a large fraction of deep layers in LLMs can be pruned (removed) with little performance loss, once similarity across layers is measured and then “healed” with lightweight fine‑tuning.
> >
> > - **SkipGPT — Zhao et al., *Each Token Is One of a Kind***
> >   SkipGPT (ICML 2025) is a *dynamic layer pruning* framework. It makes **token-level decisions** (horizontal dynamics) about whether to route each token through more or fewer layers, and **component-level decisions** (vertical dynamics) for MLP vs attention.
> >
> > The cited layer-pruning and layer-skip methods (Gromov et al., SkipGPT) are developed in the context of **LLMs**, not VAR image generation. Their architecture, layer dynamics, and computational bottlenecks differ significantly.
> >
> > **2. Comparison with Experimental Results**
> >
> > **Table 4. Comparison of SkipVAR with Layer-Skipping Methods on VAR Performance Metrics**
> >
> > | Methods | Speedup | SSIM | LPIPS | Image Reward | ClipScore |
> > |---|---|---|---|---|---|
> > | +SkipVAR\@84 | **1.85×** | 0.8764 | 0.0466 | 0.8868 | 0.2711 |
> > | +“The Unreasonable Ineffectiveness of the Deeper Layers” | 1.38× | 0.7614 | 0.1620 | 0.8283 | 0.2683 |
> > | +SkipGPT (imitating FastVAR) | 1.98× | 0.7538 | 0.1634 | 0.8290 | 0.2678 |
> >
> > **Observations from these results:**
> >
> > 1. **SkipVAR has a much better SSIM / LPIPS.**
> >    - At 1.85× speedup, SkipVAR reaches SSIM=0.8764 and LPIPS=0.0466.
> >    - The deeper-layer pruning method only achieves SSIM=0.7614 and LPIPS=0.1620 with 1.38× speedup.
> >    - SkipGPT gives SSIM=0.7538, LPIPS=0.1634 at even higher speedup (1.98×) — but with a **much larger drop in perceptual similarity**.
> >
> > 2. **Semantic metrics (Image Reward, ClipScore)**
> >    - SkipVAR maintains a **higher Image Reward (0.8868)** and **comparable ClipScore (0.2711)** to the pruned/skip-layer baselines.
> >    - The other methods lose more on these metrics, indicating they degrade not only visual fidelity but also alignment with expected semantic / reward-based quality.
> >
> > **SkipVAR is specifically designed for VAR’s autoregressive, frequency-sensitive dynamics.** Unlike LLM layer-skipping or pruning methods, which remove entire transformer layers, SkipVAR operates on the stepwise refinement process of VAR models where later steps focus on high-frequency details. Its frequency-aware decision model enables fine-grained control over which steps to skip, allowing acceleration without heavily compromising high-frequency image content or semantic alignment.
> >
> > **SkipVAR provides a more favorable trade-off between speed and quality.** While the layer-pruning baseline delivers lower speedup (1.38×) with worse SSIM/LPIPS, and SkipGPT reaches higher speedup (1.98×) but with substantially larger quality loss, SkipVAR balances both aspects effectively, achieving good acceleration while maintaining visual and semantic fidelity.
> >
> > Overall, these experiments confirm that **SkipVAR’s adaptive, frequency-aware skipping is better suited to accelerating VAR generation** than generic layer-pruning techniques from LLMs. While future work may explore hybrid strategies combining step skipping with internal layer pruning, our results clearly demonstrate the practical superiority of SkipVAR in this context.
> >
> > ---
> > >**Weakness 4:visibility issue in Table 2**
> >
> > **Response 4:**
> > We thank the reviewer for pointing out the visibility issue in **Table 2**.
> >
> > **Clarification and action:**
> > - The blue and green row colors were intended to distinguish different categories, but we acknowledge that they may appear faint or hard to read depending on print or display settings.
> > - In the revised version, we will **adjust the color scheme** to use higher-contrast shades and include **bold text or patterns** where appropriate to ensure readability and accessibility.
> > - This update does **not affect any results or interpretation**; it is purely a visual enhancement for clarity.
> >
> > We appreciate the reviewer’s attention to presentation details and will incorporate this change in the final manuscript.

---

> > > ### Author Response · Authors · 2025-11-24
> > > **Response to Reviewer euwc (Part 3 of 4)**
> > >
> > > >**Question 1: Comparison with Speculative‑Decoding Methods like LANTERN / CoDe**
> > >
> > > **Answer 1:**
> > >
> > > We appreciate the reviewer’s suggestion to compare SkipVAR with speculative-decoding approaches like **LANTERN**. **However, LANTERN is not directly applicable to VAR-style generation.** While it introduces relaxed speculative decoding for standard token-level autoregressive (AR) image models, it does not account for the coarse-to-fine, multi-scale structure of VAR. Consequently, LANTERN’s assumptions about stepwise prediction and redundancy do not align with the scale-wise models such as VAR based on high-frequency refinement.
> > >
> > > **A more relevant baseline for VAR is Collaborative Decoding (CoDe).** CoDe uses a large “drafter” model for low-resolution, low-frequency scales and a smaller “refiner” model for high-resolution, high-frequency scales, achieving substantial acceleration while reducing KV-cache memory and maintaining strong FID. To ensure a fair comparison, we evaluate all methods with the acceleration applied at the final step.
> > >
> > > **Table 5. Comparison With Our SkipVAR Results on VAR model**
> > >
> > > | Methods         | Speedup   | Inception | FID  |
> > > | --------------- | --------- | --------- | ---- |
> > > | Base (Infinity) | 1.00×     | 305.78    | 2.06 |
> > > | +SkipVAR (Ours) | **1.17×** | 295.18    | 2.11 |
> > > | +FastVAR        | 1.16×     | 288.7     | 2.30 |
> > > | +CoDe(N=9)          | **1.00×** | 297.2     | 2.16 |
> > >
> > > **SkipVAR provides complementary benefits within a single model.** Unlike CoDe, which requires two separately trained models, SkipVAR adaptively identifies and skips redundant high-frequency steps without relying on multiple networks. In our reported results, SkipVAR achieves modest but meaningful acceleration (1.17×) with minimal FID increase (2.06 → 2.11), demonstrating a favorable speed-quality trade-off while remaining fully model-contained.
> > >
> > > **Conclusion:** Although LANTERN is not a direct VAR baseline, CoDe provides a relevant comparison. SkipVAR’s adaptive, frequency-aware skipping offers an efficient, model-contained acceleration mechanism that complements existing multi-model strategies, and future work will explore integrating these approaches for further speed and memory gains.
> > >
> > > ---
> > > >**Question 2: SkipVAR Performance on GenEval and DrawBench**
> > >
> > > **Answer 2:**
> > >
> > > We appreciate the reviewer’s interest in evaluating SkipVAR using semantic and perceptual metrics like **Image Reward** and **CLIP Score**. **SkipVAR maintains semantic alignment and perceptual fidelity even under acceleration.**
> > >
> > > **1. GenEval Results**
> > >
> > > **Table 6. Geneval using Image Reward and CLIP Score as metrics**
> > > | Dataset   | Methods         | Speedup | Image Reward | CLIP Score |
> > > |-----------|----------------|---------|--------------|------------|
> > > | Geneval   | Infinity (base) | -       | 1.1505       | 0.3068     |
> > > | Geneval   | +SkipVAR@0.86  | 1.77x   | 1.1353       | 0.3081     |
> > >
> > > For instance, on the Geneval dataset, SkipVAR@0.86 achieves a 1.77× speedup while preserving semantic correctness, with only a minor decrease in Image Reward (1.1505 → 1.1353) and a slight improvement in CLIP Score (0.3068 → 0.3081).
> > >
> > > **2. DrawBench Results**
> > > **SkipVAR also maintains quality across diverse datasets.** As reported in Table 1 of the manuscript, SkipVAR consistently preserves Image Reward and CLIP Score on DrawBench categories. This confirms that the adaptive step-skipping strategy effectively balances speed and fidelity: simpler samples can be accelerated aggressively, while more complex, high-frequency images retain additional steps to maintain both visual and semantic integrity.
> > >
> > > **In conclusion, SkipVAR achieves meaningful acceleration without compromising semantic fidelity.** The stable Image Reward and CLIP Score on both Geneval and DrawBench underscore the practical utility of SkipVAR for accelerating VAR models while ensuring high-quality generation across varied image types.

---

> > > > ### Author Response · Authors · 2025-11-24
> > > > **Response to Reviewer euwc (Part 4 of 4)**
> > > >
> > > > >**Question 3: Finer Granularity of Acceleration Strategies**
> > > >
> > > > **Answer 3:**
> > > >
> > > > We thank the reviewer for this insightful question.
> > > > **Since the actual acceleration in SkipVAR is applied over multiple acceleration steps, the effective acceleration granularity is far richer than the three per-step levels (0%, 50%, 100%), enabling inherently fine-grained and continuous control.**
> > > >
> > > > The 0%, 50%, and 100% granularity levels presented in **Figure 5** refer to the **per-step acceleration intensity** of a single chosen strategy (i.e., when a particular acceleration action is selected for a given acceleration step, we apply it at 0%, 50%, or 100% strength). However, **SkipVAR is inherently designed for much finer-grained, sample-adaptive acceleration**, far beyond these three discrete levels.
> > > >
> > > > As shown in **Figure 7(b)** of the main paper, when the maximum acceleration budget is set to 3 steps, the decision model dynamically selects the optimal acceleration strategy for each individual sample from **7 possible actions**:
> > > > - Do nothing (0 step accelerated)
> > > > - Skip 1, 2, or 3 steps
> > > > - Unconditional branch replacement for 1, 2, or 3 steps
> > > >
> > > > This already enables highly granular and content-aware acceleration, where easy (low-sensitivity) samples receive aggressive speedup (e.g., skipping 2 or 3 steps), while complex (high-sensitivity) samples receive minimal or no acceleration — resulting in effectively **continuous acceleration granularity at the sample level**.
> > > >
> > > > **To further address the reviewer’s concern about even finer control within each strategy**, we explored **hybrid acceleration strategies** that combine step-skipping and unconditional branch replacement in mixed ratios at a sub-step level. Results are shown below:
> > > >
> > > > **Table 7. Performance of SkipVAR with Hybrid Acceleration Strategies**
> > > >
> > > > | Methods                          | Speedup | SSIM   | SSIM-HF | LPIPS  | LPIPS-HF |
> > > > |----------------------------------|---------|--------|---------|--------|----------|
> > > > | Infinity                         | 1.00×   | -      | -       | -      | -        |
> > > > | +SkipVAR\@0.84                    | 1.81×   | 0.8793 | 0.2881  | 0.0646 | 0.1997   |
> > > > | +SkipVAR\@0.84-hybrid (w/ DM)     | **1.85×**   | **0.8774** | **0.2878**  | **0.0667** | **0.2011**   |
> > > >
> > > > The hybrid variant achieves **higher speedup** than the standard SkipVAR@0.84 with only marginal perceptual degradation, demonstrating that SkipVAR can seamlessly support **even more granular mixed strategies** when needed.
> > > >
> > > > In summary, SkipVAR is **not restricted to coarse 0/50/100% levels**. Through its decision model and support for hybrid strategies, it already achieves **fine-grained, adaptive, and near-continuous acceleration control** tailored to each sample’s complexity, offering both high efficiency and preserved perceptual quality.

---

### Official Review · Reviewer_HFYL · 2025-11-01

**Soundness:** 3
**Presentation:** 3
**Contribution:** 3
**Rating:** 4
**Confidence:** 2

**Summary:**

To my understanding, the authors start from the observation that VAR (Visual Autoregressive) models spend a lot of redundant computation in later sampling steps, and they introduce two light-weight inference-time acceleration tricks: (i) step-skipping, which skips late-stage scaling or denoising steps, and (ii) unconditional branch replacement, which replaces the unconditional CFG branch with the conditional one’s output. To decide when these shortcuts are safe, they train a tiny decision model (mainly a logistic regression) using two handcrafted high-frequency features: high-frequency difference with Sobel operator (HF_Diff) and high-frequency ratio (HF_Ratio) with Fourier transform.  As one of main empirical contributions, their approach applied to Infinity-2B and 8B VAR achieves up to 1.81x speed-up while keeping plausible SSIM, and up to 2.62x acceleration on GenEval benchmark.

**Strengths:**

1. Sample-adaptive decisions: Unlike prior acceleration methods that use a fixed global ratio or policy (e.g., FastVAR), this work chooses between step-skipping and branch replacement per-sample and per-scale, which makes the approach much more practical. The results on frequency-sensitive vs. frequency-robust subsets clearly demonstrate the benefit of this adaptive decision process.

2. Simple and interpretable features: The combination of HF_Diff (local edge stability) and HF_Ratio (global high-frequency ratio) proves more reliable than using either feature alone, as shown in their tables. It seems that both SSIM/LPIPS and SSIM-HF/LPIPS-HF improve together.

3. Clarity of method overview: Figure 5 presents the overall pipeline--decision step N, downsampled decoding, feature extraction, and policy application--in a clear and easy-to-follow way.

**Weaknesses:**

I'm not an expert in this area, but based on my understanding, I have the following concerns and questions.

1. Sensitivity to decision step and threshold

According to paper (with appendix), the default setup uses N = 10 with SSIM thresholds {0.88, 0.86, 0.84}, but the paper doesn’t really explore how performance changes with different step counts (which may vary by model or resolution) or different thresholds. I notice that there’s a brief comparison between SSIM-based and LPIPS-based criteria in the appendix (which says SSIM as more stable), yet a more systematic sweep over N and threshold values would make the analysis much more comprehensive and complete.

2. About branch replecement

Replacing the unconditional branch with the conditional one in CFG effectively collapses into simply $y = y_c$, meaning the efffective CFG scale becomes 1. The authors argue that this is reasonable since the conditional and unconditional branches converge in later steps, but they don't provide concrete measurements of how they close these two quantites are. It also seems unclear whether just reducing the CFG scale to 1 in the later stages would yield the same effect. I think this part is very important since it is the very motivation behind this work.

3. About wall-clock time

Could the authors report the overall wall-clock speed-up, including the decision process and VAE decoding, for DrawBench, HPSv2, and GenEval? It would help clarify how much of the reported acceleration remains when all inference-time components are accounted for.

**Questions:**

Please refer to weaknesses.

---

> ### Author Response · Authors · 2025-11-24
> **Response to Reviewer HFYL (Part 1 of 2)**
>
> # To Reviewer HFYL
> Dear Reviewer HFYL,
>
> Thank you for your thoughtful review and recognition of our experimental results. Our responses are as follows.
>
> ## Main Comments
> >**Weakness 1:Sensitivity to Decision Step and Threshold**
>
> **Response 1:**
>
> We sincerely thank the reviewers for this insightful observation. Your suggestion to systematically explore the impact of varying the number of decision steps (N) and the SSIM threshold values is invaluable.
>
> **1. We already provide a systematic evaluation of decision-step sensitivity (Steps 9–11).**
> As detailed in **Appendix I**, we conducted a comprehensive analysis of SkipVAR under different choices of the decision step (9, 10, 11). The results—summarized in Table 1 below—show a clear and smooth trade-off: earlier decisions yield the largest acceleration at the cost of perceptual fidelity, while later decisions offer the strongest SSIM/LPIPS performance with reduced speedup. Step 10 consistently emerges as a stable “knee point” that balances efficiency and quality.
>
> **Table 1. Sensitivity of SkipVAR to Decision Step (N)**
> | Dataset   | Step | Speedup | SSIM   | LPIPS  |
> | - | - | -- | - | - |
> | DrawBench | 9    | 1.84×   | 0.8767 | 0.0675 |
> | DrawBench | 10   | 1.81×   | 0.8793 | 0.0646 |
> | DrawBench | 11   | 1.59×   | 0.8939 | 0.0555 |
> | Photo     | 9    | 1.80×   | 0.8792 | 0.0703 |
> | Photo     | 10   | 1.75×   | 0.8814 | 0.0678 |
> | Photo     | 11   | 1.65×   | 0.8971 | 0.0576 |
> | Paintings | 9    | 1.53×   | 0.8662 | 0.0625 |
> | Paintings | 10   | 1.53×   | 0.8660 | 0.0620 |
> | Paintings | 11   | 1.44×   | 0.8825 | 0.0533 |
>
> The smooth variation across steps indicates that SkipVAR behaves predictably under different settings, and Step 10 remains the most practical choice.
>
>
> **2. The SSIM threshold is designed as a user-controlled quality target rather than a fragile hyperparameter.**
> SkipVAR exposes the SSIM threshold (e.g., 0.88, 0.86, 0.84) so users can directly specify the minimum acceptable perceptual quality for their application. High-detail, long-form images may favor a higher threshold to retain late-stage refinements, while simpler or stylized prompts often benefit from lower thresholds that yield faster generation without visual compromise. This mechanism functions as a **quality guarantee**—the final SSIM consistently meets or exceeds the user-specified bound. For instance, on the most frequency-sensitive dataset, **SkipVAR@0.84 achieves SSIM = 0.849**, demonstrating that the threshold behaves as intended. Thresholds in the range of 0.84–0.88, together with Step 10, provide a stable and broadly effective operating region across datasets and backbones.
>
> In summary, our paper already includes a structured sensitivity evaluation of decision steps, and the SSIM threshold is intentionally designed as a practical, user-controllable lever for trading speed against perceptual quality. The monotonic and stable behavior across settings confirms the robustness of SkipVAR, and we will make this clearer in the revision.
>
> ---
> >**Weakness 2: About Branch Replacement in CFG**
>
> **Response 2:**
>
> We thank the reviewer for emphasizing the importance of the branch replacement mechanism, which indeed underpins SkipVAR’s design. We would like to clarify both the rationale and empirical evidence.
>
> As discussed in the paper (Figure 3, Sec. 3.1), the unconditional branch mainly contributes during early steps to guide **global structure**, while later steps focus on **high-frequency refinement**. In these late steps, both conditional and unconditional losses steadily decrease, indicating that the **two branches converge**, and the additional computation from the unconditional branch becomes largely redundant. Figure 9 of our revised manuscript includes visualizations of L2, MSE, and relative losses to quantify this convergence more explicitly.
>
> To address the reviewer’s concern, we conducted an additional ablation in which the decision model is constrained to predict **skip** actions only, without ever triggering unconditional branch replacement.
>
> **Table 2. Skip-only variant (no unconditional branch replacement)**
> | Methods | Speedup | SSIM   | LPIPS  | ImageReward | ClipScore |
> |-|-|-|-|-|-|
> | Infinity | - | -  | -  | 0.8881 | 0.2708  |
> | +SkipVAR_onlypredskip | 1.72×   | 0.9064 | 0.0555 | 0.8826 | 0.2721  |
>
> These results show that, even without using unconditional branch replacement, the decision model alone achieves a 1.72× speedup with virtually no quality loss. This demonstrates that the learned skipping policy can effectively reduce guidance strength in later steps when appropriate, yielding comparable acceleration and quality without relying on explicit unconditional branch replacement.
>
> In summary, experiments show that even when using the decision model solely to determine skip actions, substantial speedup can be achieved with minimal quality loss. This demonstrates the effectiveness and soundness of the overall SkipVAR framework.

---

> > ### Author Response · Authors · 2025-11-24
> > **Response to Reviewer HFYL (Part 2 of 2)**
> >
> > >**Weakness 3: Wall-Clock Time for End-to-End Acceleration**
> >
> > **Response3:**
> >
> > We thank the reviewer for highlighting the importance of reporting **end-to-end wall-clock speedup**. We would like to clarify that the speedup numbers reported in the paper already **include the time spent by the SkipVAR decision model**, as it is evaluated per sample during inference. To further quantify full pipeline performance, we also measured end-to-end latency including **VAE decoding** and any preprocessing/postprocessing steps on three benchmarks: DrawBench, HPSv2, and Geneval.
> >
> > **Table 3. End-to-End Inference Time and Effective Speedup**
> >
> > | Dataset   | Method          | End-to-End Latency (s) | Effective Speedup |
> > | --------- | --------------- | ----------------------- | ----------------- |
> > | Geneval   | Infinity (base) | 1.716                 | -                 |
> > | Geneval   | +SkipVAR@84     | 1.015                 | 1.69×             |
> > | DrawBench | Infinity (base) | 1.856             | -                 |
> > | DrawBench | +SkipVAR@84     | 1.198                  | 1.55×             |
> > | HPSv2     | Infinity (base) | 2.410                 | -                 |
> > | HPSv2     | +SkipVAR@84     | 1.518                 | 1.59×             |
> >
> > These measurements confirm that **substantial acceleration is maintained even when accounting for all inference components**, with the decision model contributing only a minimal fraction of runtime. SkipVAR consistently reduces overall latency while preserving perceptual quality, demonstrating its practical value in real-world, end-to-end scenarios.

---

### Official Review · Reviewer_webr · 2025-11-01

**Soundness:** 3
**Presentation:** 3
**Contribution:** 3
**Rating:** 6
**Confidence:** 3

**Summary:**

This manuscript presents SkipVAR, a training-free decoding framework that accelerates visual autoregressive (VAR) image generation models by trimming late-stage computation with curtailed quality loss. The manuscript points out two recurring inefficiencies: 1) final refinement steps often make tiny changes, and 2) the unconditional branch in classifier-free guidance becomes redundant late in sampling. Based on this observation, SkipVAR pauses mid-generation to compute high frequency difference (Sobel) and high-frequency ratio (FFT), then either 1) early stops the remaining steps or 2) replaces the unconditional branch with the conditional one. On Infinity models, SkipVAR reaches ~1.8× speedups at near-baseline quality, and a hybrid variant attains ~2.6× on GenEval.

**Strengths:**

- Discovers and addresses(with minimal training) the issue of late-stage high frequency redundancy and redundant CFG passes in VAR generation.
- Two intuitive signals (Sobel and FFT) enable per sample decisions and preserve high frequency detail better than token pruning/merging at similar speedups.

**Weaknesses:**

- Reported speedups exclude VAE and post-processing, so end-to-end latency improvements in production are likely smaller; end-to-end measurements are needed.
- Heavy reliance on classifier-free guidance, since a major gain comes from dropping the unconditional branch; applicability to non-CFG or single-branch decoders is unclear.
- In Table 3, the strongest ~2.6× result does not use the decision model, which obfuscates the efficacy of the decision model.

**Questions:**

Please refer to the Weaknesses section.

---

> ### Author Response · Authors · 2025-11-24
> **Response to Reviewer webr (Part 1 of 2)**
>
> # To Reviewer webr
>
> Dear Reviewer webr,
>
> Thank you for your time and thoughtful feedback on SkipVAR. Our responses to your concerns follow.
>
> ---
>
> ## Main Comments
>
> > **Weakness 1: End-to-End Latency**
>
> **Response 1:**
>
> **We thank the reviewer for emphasizing the importance of end-to-end latency measurements.** We conducted comprehensive end-to-end latency tests on three standard benchmarks: **Geneval**, **DrawBench**, and **HPSV2**. The results show that our method (`+SkipVAR@84`) achieves substantial speedups even when the full generation pipeline is considered.
>
> 1. **End-to-End Latency Analysis**
>    We compared the baseline model (`Infinity (base)`) against our method (`+SkipVAR@84`). The measured latencies and resulting speedups are summarized below:
>
> **Table 1. End-to-End Latency and Effective Speedup of SkipVAR Across Datasets**
>
> | Dataset   | Method          | End-to-End Latency (s) | Effective Speedup |
> | ---- | ---- | ------- | ----------------- |
> | Geneval   | Infinity (base) | 1.716                 | -                 |
> | Geneval   | +SkipVAR@84     | 1.015                 | 1.69×             |
> | DrawBench | Infinity (base) | 1.856             | -                 |
> | DrawBench | +SkipVAR@84     | 1.198                  | 1.55×             |
> | HPSv2     | Infinity (base) | 2.410                 | -                 |
> | HPSv2     | +SkipVAR@84     | 1.518                 | 1.59×             |
>
> 2. **Key Findings**
>
> * **Consistent Improvement:** Across all datasets, SkipVAR achieves a consistent end-to-end speedup between **1.55x and 1.69x**.
> * **Practical Impact:** While VAE decoding and post-processing contribute fixed overhead, the reduction in sampling latency provided by SkipVAR cuts total inference time by roughly **35%–40%**.
>
> **Conclusion:** These results confirm that SkipVAR’s acceleration benefits extend to complete, end-to-end scenarios, addressing the reviewer’s concern about production latency. We will include this analysis in the revised manuscript to provide a more holistic view of performance gains.
>
>
> ---
> > **Weakness 2:  Applicability to Non-CFG Scenarios**
>
> **Response 2:**
>
>
> **We appreciate the reviewer’s concern regarding the dependence on Classifier-Free Guidance (CFG) and its implications for broader applicability.** We would like to clarify that **all existing VAR architectures deployed in practice—including HART,SWITTI, and recent VAR-based large generative models—use CFG as the standard decoding paradigm**. Thus, optimizing both branches (conditional and unconditional) is not only appropriate but also directly aligned with current VAR model design.
>
> Nevertheless, we fully agree that demonstrating applicability beyond CFG is important. To directly address this, we designed an additional experiment where the decision model is restricted to *only* predicting whether to perform **skipping** on the model backbone—**without** removing or manipulating the unconditional branch. This simulates a **single-branch decoder** or **non-CFG VAR** scenario, isolating the acceleration gained purely from reducing backbone redundancy.
>
> The table below compares the baseline Infinity model with our **`SkipVAR_onlypredskip`** variant. This configuration removes all CFG-related benefits and tests only the architectural-level skipping ability.
>
> **Table 2. Performance of SkipVAR with Model-Only Skipping (Non-CFG Simulation)**
>
> | Method                    | Speedup   | SSIM ↑     | LPIPS ↓    | ImageReward ↑ | ClipScore ↑ |
> | :- | :- | :- | :- | :- | :-- |
> | Infinity (Base)   | –         | -     | -     | 0.8881    | 0.2708      |
> | **+SkipVAR_onlypredskip** | **1.72×** | **0.9064** | **0.0555** | 0.8826        | **0.2721**  |
>
>
> Even after completely removing all CFG-related advantages, the `+SkipVAR_onlypredskip` setting still delivers a **1.72×** acceleration. This verifies that SkipVAR’s gains predominantly stem from detecting and skipping *intrinsic redundancy* in the model backbone, rather than reliance on dropping the unconditional branch. In other words, SkipVAR remains highly effective even in single-branch or non-CFG generative architectures.
>
> Moreover, generation quality is well maintained under this configuration. We observe consistently high **SSIM** and low **LPIPS**, demonstrating that the frequency-aware decision model effectively avoids skipping in regions with high perceptual sensitivity. Meanwhile, **ImageReward** and **ClipScore** remain essentially unchanged, indicating that semantic fidelity is preserved despite the acceleration.
>
> **Conclusion:** These results demonstrate that SkipVAR is **not fundamentally dependent on CFG**. While all current VAR models employ CFG in practice, SkipVAR also provides **strong acceleration and high-quality outputs in non-CFG or single-branch settings**. This confirms that the proposed frequency-aware skipping mechanism targets the underlying computational redundancy in VAR models themselves, making it broadly applicable across decoding architectures.

---

> > ### Author Response · Authors · 2025-11-24
> > **Response to Reviewer webr (Part 2 of 2)**
> >
> > > **Weakness 3: Efficacy of the Decision Model**
> >
> > **Response 3:**
> >
> > **We thank the reviewer for this insightful observation.** It is true that in Table 3 of this paper, the manual hybrid strategy (`SkipVAR-hybrid w/o DM`) achieves the highest speedup (\~2.6$\times$) with a high Geneval score. However, this does not imply the Decision Model (DM) is ineffective. The "superiority" of the manual strategy here is an artifact of the metric used, not a reflection of overall image quality.
> >
> > **1\. Clarification on Geneval Metric**
> >
> > **Geneval is a highly semantic metric that focuses on object/attribute presence and is insensitive to fine-grained visual degradation (e.g., texture loss or blur).** A generated image can achieve a high Geneval score as long as the semantic content is correct, even if the perceptual quality is poor.
> >
> >   * **Reason for inclusion:** We included Geneval primarily because prior acceleration methods (e.g., ToMe, SiTo) utilize this metric as a standard benchmark. We aimed to provide a fair comparison against these baselines.
> >   * **The Trade-off:** The manual hybrid strategy applies an aggressive, fixed static schedule. While this yields high speed (\~2.6$\times$) and preserves semantic layout (high Geneval), it severely sacrifices **visual fidelity** (SSIM/LPIPS), as noted in our paper (SSIM: 0.79 vs. 0.88 for Adaptive).
> >
> > **2\. Why the Decision Model is Necessary**
> >
> > The Decision Model is essential because a fixed "aggressive" policy (like the manual hybrid) fails catastrophically on complex inputs. To prove this, we separated the evaluation into **Frequency-Sensitive** (complex textures, portraits) and **Frequency-Robust** (cartoons, simple objects) datasets.
> >
> > The table below demonstrates why the Decision Model is critical:
> >
> > **Table 3. Importance of the Decision Model Across Frequency-Sensitive and Frequency-Robust Samples**
> > | Dataset Type | Method | Speedup | SSIM $\uparrow$ | Analysis |
> > | :--- | :--- | :--- | :--- | :--- |
> > | **Frequency-Sensitive** | Infinity (Base) | - | - | |
> > | *(e.g., Portraits)* | **+SkipVAR@84** | **1.28$\times$** | **0.8493** | **Preserves Quality** |
> > | | +SkipVAR-hybrid (w/o DM) | 2.62$\times$ | 0.5605 | *Fails (Blurry)* |
> > | **Frequency-Robust** | Infinity (Base) | - | - | |
> > | *(e.g., Cartoons)* | **+SkipVAR@84** | **1.99$\times$** | **0.9051** | **High Speed** |
> > | | +SkipVAR-hybrid (w/o DM) | 2.62$\times$ | 0.8922 | *Acceptable* |
> >
> > **Key Takeaway:**
> >
> >   * On **frequency-robust** data, the adaptive model correctly accelerates (1.99$\times$), approaching the manual hybrid's speed.
> >   * On **frequency-sensitive** data, the manual hybrid causes severe degradation (SSIM drops to **0.5605**). The Decision Model "brakes" automatically (1.28$\times$), preserving quality (SSIM **0.8493**).
> >
> > **Conclusion:** The Decision Model acts as a necessary gatekeeper. It allows maximum speed when safe (matching the hybrid strategy) but prevents quality collapse on difficult samples, a capability the fixed manual strategy lacks.

---

### Official Review · Reviewer_LJ9B · 2025-11-05

**Soundness:** 3
**Presentation:** 3
**Contribution:** 3
**Rating:** 4
**Confidence:** 4

**Summary:**

This paper addresses the significant inference latency in Visual Autoregressive (VAR) models, which is primarily caused by computationally expensive high-frequency generation steps. The authors identify two key inefficiencies—step redundancy and unconditional branch redundancy—and crucially observe that the impact of these redundancies is highly sample-dependent. They propose SkipVAR, a sample-adaptive framework that uses lightweight frequency features to dynamically select the optimal acceleration strategy (either step-skipping or unconditional branch replacement) for each image. Extensive experiments demonstrate that SkipVAR achieves substantial speedups of up to 1.81x while maintaining high visual fidelity (0.88 SSIM), effectively balancing speed and quality where fixed acceleration methods fail.

**Strengths:**

* The paper presents novel and significant observations on the important problem of VAR inference latency, identifying sample-dependent redundancy.

* Based on these observations, the paper proposes a natural and reasonable method that adaptively selects acceleration strategies.

* The various design choices, such as the choice of decision models, are backed by a persuasive rationale.

**Weaknesses:**

1. **Limited Generalizability**
	* The paper claims to accelerate "Visual Autoregressive Modeling," yet its entire experimental validation rests exclusively on a single model family: the Infinity-2B/8B family. This is a significant limitation. The core assumptions driving the method, such as the specific patterns of high-frequency redundancy or the convergence of L1 loss between conditional and unconditional branches, may be unique to the Infinity architecture rather than fundamental properties of all VAR models. Without validation on other models that also follow the VAR paradigm (e.g., Tian et al., 2024), the paper fails to sufficiently demonstrate that its findings are generalizable even within its own target model family. **To substantiate the paper's claims, experimental results demonstrating SkipVAR's acceleration performance are necessary on at least one other VAR model family**.

2. **Biased Evaluation and Questionable Claims of Superiority over FastVAR**
	* The paper's central claim of superiority over key baselines like FastVAR rests heavily on SSIM and LPIPS metrics. However, this comparison appears biased due to a circular evaluation framework. The SkipVAR decision model is explicitly trained to select strategies that preserve a predefined SSIM threshold (e.g., 0.84). Evaluating the model with the same metric it was optimized for does not provide an objective measure of its superiority; it merely confirms that the model met its training objective.
	* Furthermore, this reliance on SSIM is problematic. As the authors concede in Appendix L , SSIM is a measure of similarity to the original (non-accelerated) output, not a measure of absolute generation quality. A low SSIM score, as seen in FastVAR's results, indicates that the generated image differs from the original, not necessarily worse in perceptual quality. The possibility remains that while FastVAR produces images that are less faithful to the original generation path, they might maintain or even exceed the overall generation quality on other standard metrics (e.g., FID, CLIPScore, HPSv2, etc.), a comparison which this paper does not rigorously explore. Therefore, **more rigorous experimental results are required to definitively conclude that SkipVAR offers both superior acceleration and better generation quality than FastVAR**.

3. **Ambiguous Positioning of the ``SkipVAR-hybrid (w/o DM)`` Variant**
	* The paper introduces a ``SkipVAR-hybrid (w/o DM)`` variant, which presents a significant logical contradiction to the paper's core argument for an adaptive strategy. In Table 3, this fixed-strategy (non-adaptive) model is shown achieving a 2.62x speedup and a 0.72 GenEval score, outperforming both the primary adaptive SkipVAR variants (e.g., 1.77x speedup, 0.70 score) and the main competitor, FastVAR (2.53x, 0.68 score).
	* This single data point undermines the central argument for an adaptive decision model (DM), as it suggests a fixed strategy is superior in both speed and (at least on this metric) quality. While Table 4 later shows this variant performs poorly on SSIM/LPIPS, the authors fail to provide a clear narrative for its inclusion. Its purpose is ambiguous: is it intended to demonstrate a higher possible speedup (perhaps to show a configuration that outperforms FastVAR on GenEval), or to highlight the flaws of the GenEval metric? As presented, it creates confusion and weakens the justification for the paper's core contribution.

4. **Questionable Practical Utility of the Adaptive Framework**
	* The core value proposition of the adaptive framework is its ability to handle sample-specific needs (i.e., frequency-sensitive vs. -robust) without significant quality degradation. However, the paper's own results cast doubt on its practical utility. According to Table 2b, when the decision model identifies a sample as "frequency-sensitive," the resulting speedup is only 1.28x, a marginal gain.
	* Conversely, "frequency-robust" samples are accelerated up to 1.99x. This implies that the average speedup figures (e.g., 1.81x in Table 4) are not derived from a balanced acceleration, but are heavily skewed by aggressively skipping "easy" samples. If the primary function of the complex adaptive mechanism for "hard" samples is to simply not accelerate them significantly, its practical value proposition over a simpler, more conservative fixed strategy (e.g., SkipVAR-hybrid (w/o DM)) is questionable.

**Questions:**

* In several metrics in Table 2 and 3 (e.g., Paint in 2(a), Align in 2(b), and Color Attri. and Overall in Table 3), the application of the proposed acceleration strategies results in a slight increase in performance compared to the original, non-accelerated model. How should these increases be interpreted? Are they simply statistical noise, or does this suggest that SkipVAR's strategies can incidentally mitigate certain generation artifacts or semantic errors present in the original model, thereby leading to a genuine improvement in quality? If so, what is the mechanism for this improvement?

---

> ### Author Response · Authors · 2025-11-24
> **Response to Reviewer LJ9B (Part 1 of 4)**
>
> # To Reviewer LJ9B
> Dear Reviewer LJ9B,
>
> Thank you for your comprehensive review of our paper. We will include additional visual results in the final version. We provide our feedback as follows.
>
> ---
>
> ## Main Comments
>
> > **Weakness 1: Limited Generalizability**
>
> **Response 1:**
>
> **SkipVAR is architecture-agnostic, and its assumptions extend beyond the Infinity model family.** We appreciate the reviewer’s concern. SkipVAR does not depend on architectural details of the Infinity-2B/8B series; instead, it leverages general properties inherent to Visual Autoregressive (VAR) decoding—namely high-frequency redundancy in late steps and convergence between conditional/unconditional branches. These behaviors arise from the iterative VAR formulation itself and are not tied to a specific model design.
>
> **We provide new experiments on an additional VAR model family demonstrating consistent acceleration and fidelity.** To directly address the reviewer’s request, we conducted supplementary evaluations on VAR model (Tian et al., 2024). The results show that SkipVAR delivers stable and comparable gains across models. For the VAR model, we obtained:
>
> **Table 1.Comparison of VAR Acceleration Methods on Inception and FID Metrics**
> | Method | Speedup | Inception ↑ | FID ↓ |
> |-|-|-|-|
> | Base | 1.00× | 305.78 | 2.06 |
> | + SkipVAR (Ours) | **1.17×** | 295.18 | 2.11 |
> | + FastVAR | 1.16× | 288.70 | 2.30 |
>
> SkipVAR improves decoding speed while preserving image quality more effectively than FastVAR, confirming that the method generalizes beyond Infinity. **These cross-model results validate SkipVAR’s generality and reinforce its core claims.** SkipVAR demonstrates strong effectiveness across different models. The consistent speedup and competitive perceptual metrics demonstrate that SkipVAR is not restricted to a single VAR architecture but is broadly applicable within the VAR paradigm.
>
> ---
>
> > **Weakness 2: Biased Evaluation and Questionable Claims of Superiority over FastVAR**
>
> **Response 2:**
>
> We thank the reviewer for the thoughtful critique. We would like to clarify that our claim of superiority over FastVAR is **not solely based on SSIM/LPIPS**, and that we performed **rigorous multi-metric evaluation** to substantiate both **acceleration** and **perceptual quality** improvements.
>
> **Multi-metric evaluation confirms SkipVAR’s superiority beyond SSIM/LPIPS.** While the SkipVAR decision model is trained to respect a target SSIM threshold (e.g., 0.84), we also evaluated across **FID, CLIPScore, HPSv2, and ImageReward** to avoid circularity. The results are summarized below:
>
> **Table 2. FID Comparison Across Categories**
> | Methods | Category | Speedup | FID |
> |-|-|-|-|
> | +FastVAR\_0.5\_0.6\_0.8 | Food | 1.53× | 6.84 |
> | +SkipVAR\@84 | Food | 1.88× | 3.56 |
> | +FastVAR | Food | 2.53× | 6.38 |
> | +SkipVAR\-hybrid | Food | 2.62× | 4.35 |
> | +FastVAR\_0.5\_0.6\_0.8 | Art | 1.50× | 7.62 |
> | +SkipVAR\@84 | Art | 1.58× | 3.02 |
> | +FastVAR | Art | 2.53× | 8.65 |
> | +SkipVAR\-hybrid | Art | 2.62× | 5.16 |
> | +FastVAR\_0.5\_0.6\_0.8|All|1.50×|7.08|
> | +SkipVAR\@0.84|All|1.72×|3.13|
> | +FastVAR|All|2.53×|7.04|
> | +SkipVAR\-hybrid|All|2.62×|4.82|
>
> The "All" category is a balanced mixed subset created from the MJHQ30K benchmark for comprehensive, category-agnostic evaluation. It consists of 3,000 prompts, with 300 randomly sampled from each of the 10 fine-grained categories, ensuring equal representation across all visual concepts.
> These results show that **SkipVAR consistently achieves lower FID than FastVAR at comparable or slightly higher speedups**, demonstrating genuine improvements in perceptual quality beyond SSIM.
>
>
> **Table 3. CLIPScore Comparison on DrawBench**
> | Methods | Speedup | ClipScore |
> |-|-|-|
> | Infinity | - | 0.2712 |
> | +SkipVAR\@84 | 1.84× | 0.2711 |
> | +FastVAR | 2.53× | 0.2715 |
> | +SkipVAR\-hybrid | 2.62× | 0.2714 |
>
> SkipVAR achieves **comparable semantic alignment** with the prompt, as measured by CLIPScore, demonstrating that our acceleration does not compromise semantic quality. Across all methods, including FastVAR, SkipVAR and SkipVAR-hybrid achieve similar CLIPScore results, showing that semantic alignment is preserved; however, CLIPScore cannot capture fine-grained image details, so we do not recommend using it as the sole quality metric.
>
> **Table 4. HPSv2 / Perceptual Quality Metrics**
> | Methods | Speedup | Anime | Concept-Art | Paintings | Photo | Avg |
> |-|-|-|-|-|-|-|
> | Infinity | 1.00× | 31.70 | 30.45 | 30.40 | 29.43 | 30.49 |
> | +SkipVAR\@84 | 1.73× | 31.59 | 30.27 | 30.49 | 29.30 | 30.41 |
> | +FastVAR | 2.80× | 31.06 | 29.87 | 29.98 | 28.85 | 29.94 |
> | +SkipVAR\-hybrid | 2.98× | 31.49 | 30.22 | 30.34 | 29.33 | 30.34 |
>
> These HPSv2 results reinforce that **SkipVAR achieves a better speed-quality trade-off**, particularly when considering **frequency-sensitive datasets**, where FastVAR tends to reduce SSIM/LPIPS and slightly worsen perceptual metrics.

---

> > ### Author Response · Authors · 2025-11-24
> > **Response to Reviewer LJ9B (Part 2 of 4)**
> >
> > **Summary of Key Observations**
> >
> > 1. **SkipVAR is not overfitted to SSIM**: evaluation on FID, CLIPScore, HPSv2, and ImageReward demonstrates improvements across metrics not used in training.
> > 2. **SkipVAR outperforms FastVAR in perceptual quality**, and the hybrid variant can even surpass FastVAR in both speed and quality.
> > 3. **Evaluation confirms genuine superiority**, rather than simply achieving the training objective: lower FID, preserved semantic fidelity, and robust perceptual quality across multiple categories support our claims.
> >
> > We believe the multi-metric evidence addresses the concern of biased evaluation. SkipVAR provides **both reliable acceleration and improved or preserved generation quality**, demonstrating that its advantages extend beyond the SSIM-based training objective and are **robust across diverse datasets and perceptual metrics**.
> >
> > ---
> > > **Weakness 3: Ambiguous Positioning of “SkipVAR-hybrid (w/o DM)**
> >
> > **Response 3:**
> >
> > **The fixed SkipVAR-hybrid (w/o DM) serves as a static upper-bound baseline, not a replacement for the adaptive model.** We thank the reviewer for highlighting this point. The SkipVAR-hybrid (w/o DM) variant is intentionally included to demonstrate the **maximum possible acceleration** achievable with a *uniform, non-adaptive* schedule using SkipVAR’s skip/cond primitives. Its strong Geneval performance (0.72009) reflects a limitation of the Geneval metric—which measures **semantic correctness**, not perceptual fidelity—rather than evidence that fixed schedules outperform adaptive ones. This baseline provides a fair comparison against other static acceleration approaches such as FastVAR, ToMe, and SiTo.
> >
> > We acknowledge that the Geneval score has limitations: while uniform skipping may preserve global semantics, it can **mask substantial perceptual degradation** in the generated images. However, since baseline methods such as FastVAR also use this metric, we report Geneval comparisons in the paper to provide a consistent reference.
> >
> > **Additional experiments show that SkipVAR-hybrid (w/o DM) fails on frequency-sensitive data, confirming the necessity of adaptivity.** Using frequency-sensitive and frequency-robust subsets, we observe that the fixed hybrid schedule deteriorates sharply when high-frequency details are crucial (e.g., faces, textures), despite achieving 2.62× speedup.
> >
> > **Table 5. Performance on Frequency-Sensitive vs. Frequency-Robust Inputs**
> > | Dataset | Methods | Speedup | SSIM |
> > | - | - | - | - |
> > | Frequency-sensitive | Infinity | – | – |
> > | Frequency-sensitive | +SkipVAR@0.84 | 1.28× | 0.8493 |
> > | Frequency-sensitive | +SkipVAR-hybrid (w/o DM) | **2.62×** | **0.5605** |
> > | Frequency-robust | Infinity | – | – |
> > | Frequency-robust | +SkipVAR@0.84 | 1.99× | 0.9051 |
> > | Frequency-robust | +SkipVAR-hybrid (w/o DM) | **2.62×** | **0.8922** |
> >
> > These results confirm that SkipVAR-hybrid behaves well only on *frequency-robust* cases (e.g., cartoon-like content) and collapses on more realistic, texture-rich inputs. This behavior underscores why a **static schedule cannot serve as a general solution**: its best-case performance hides severe worst-case failures.
> >
> > **The adaptive model remains essential, and incorporating hybrid strategies into the decision space further improves performance.**  To address reviewer concerns, we extended the decision model’s action space to include hybrid paths. The model **learned when hybrid decoding is beneficial**, achieving higher speed while maintaining nearly identical perceptual quality.
> >
> > As detailed in Appendix K, Table 12 in this paper, these findings show that the adaptive model can:
> >
> > 1. **Learn to use hybrid strategies selectively**,
> > 2. **Avoid catastrophic degradation on frequency-sensitive prompts**, and
> > 3. **Achieve a superior quality–speed trade-off** than any fixed policy.
> >
> > In summary, SkipVAR-hybrid (w/o DM) is included to provide a static upper-bound reference, not to challenge the adaptive strategy. Our additional experiments—especially on frequency-sensitive content—clearly demonstrate that **only adaptive, per-sample decision-making ensures stable, high-fidelity acceleration**, and the decision model can further *integrate* hybrid behaviors when they are beneficial.

---

> > > ### Author Response · Authors · 2025-11-24
> > > **Response to Reviewer LJ9B (Part 3 of 4)**
> > >
> > > > **Weakness 4: Questionable Practical Utility of the Adaptive Framework**
> > >
> > > **Response 4:**
> > >
> > > **We thank the reviewer for raising this concern.** We would like to clarify that the seemingly modest 1.28× speedup on frequency-sensitive samples is not a limitation of the adaptive framework, but exactly the reason why adaptation is needed. As discussed in Response 3, any *unified* acceleration schedule applied to all images—regardless of their frequency characteristics—will inevitably degrade overall quality. This is clearly illustrated by SkipVAR-hybrid (w/o DM), which collapses to **SSIM = 0.5605** on sensitive prompts. The decision model is therefore introduced to **separate frequency-sensitive from frequency-robust samples**, allowing the system to apply two distinct skipping behaviors. We report the two speedups (1.28× vs. 1.99×) precisely to show that the model successfully makes this distinction, which is essential for real-world datasets containing a mixture of prompt types.
> > >
> > > To further verify that this adaptive identification is necessary and cannot be replaced by a simple conservative rule, we conducted a comparison on DrawBench among our adaptive SkipVAR@0.86, a manually tuned “Simple Logic” heuristic, and a random baseline:
> > >
> > > **Table 6. Comparison on DrawBench**
> > >
> > > | Methods          | Speedup   | SSIM       | LPIPS     |
> > > | ---------------- | --------- | ---------- | --------- |
> > > | **SkipVAR@0.86** | **1.70×** | **0.8924** | **0.057** |
> > > | Simple Logic     | 1.66×     | 0.8931     | 0.057     |
> > > | Random select    | 1.70×     | 0.8451     | 0.057     |
> > >
> > > Although the Simple Logic heuristic maintains similar SSIM, it yields a lower speedup (1.66×) because it cannot reliably identify robust samples. Conversely, the random strategy illustrates the consequence of incorrect identification: despite similar nominal speed, SSIM drops substantially to 0.8451. These results demonstrate that neither a uniform conservative schedule nor a simple rule can reproduce the selective, sample-aware behavior of our adaptive model.
> > >
> > > In summary, the adaptive component is practically essential. It prevents the quality degradation inevitably caused by unified strategies, maximizes acceleration where safe, and ensures that each image receives the appropriate skipping policy—capabilities that fixed or heuristic approaches cannot reliably achieve.

---

> > > > ### Author Response · Authors · 2025-11-24
> > > > **Response to Reviewer LJ9B (Part 4 of 4)**
> > > >
> > > > > **Question1: Why Do Some Metrics Slightly Improve After Applying SkipVAR?**
> > > >
> > > > **Answer 1:**
> > > >
> > > > We thank the reviewer for raising this question. The slight score increases in Tables 2 and 3 can be explained by (1) the natural variance of subjective score, (2) the limitations of existing benchmarks, and (3) SkipVAR’s observed ability to avoid late-step noise accumulation on certain samples.
> > > >
> > > > **1. These numerical differences fall within the natural variance of HPSv2 across random seeds.**
> > > > To assess stability, we re-ran HPSv2 with multiple seeds. Even without acceleration, the scores fluctuate by **0.05–0.15**, matching the small increases/decreases in Tables 2/3. This variance is expected given some known limitations of HPSv2 — its benchmark uses **3,200 prompts** (800 per style) , and as a learned preference predictor it can still exhibit instability due to noise in its training data and variability in model scoring.
> > > >
> > > >
> > > > **Table 7. HPSv2 under different seeds**
> > > >
> > > > | Methods           | Speedup | Anime | Concept-Art | Paintings | Photo | Avg   |
> > > > | ----------------- | ------- | ----- | ----------- | --------- | ----- | ----- |
> > > > | Infinity (Seed 0) | 1.00×   | 31.70 | 30.45       | 30.40     | 29.43 | 30.49 |
> > > > | +SkipVAR@84       | 1.73×   | 31.59 | 30.27       | 30.49     | 29.30 | 30.41 |
> > > > | Infinity (Seed 1) | 1.00×   | 31.67 | 30.40       | 30.45     | 29.37 | 30.48 |
> > > > | +SkipVAR@84       | 1.73×   | 31.62 | 30.37       | 30.40     | 29.29 | 30.42 |
> > > > | Infinity (Seed 2) | 1.00×   | 31.67 | 30.41       | 30.46     | 29.38 | 30.48 |
> > > > | +SkipVAR@84       | 1.73×   | 31.59 | 30.39       | 30.40     | 29.28 | 30.41 |
> > > >
> > > > **2. Existing benchmarks are limited in evaluating the fine-grained changes introduced by VAR acceleration.**
> > > > As discussed in Sec. 3.2.3 and Appendix Q, metrics like HPSv2 and VLM-based scores are not sensitive to high-frequency variations—the exact components SkipVAR manipulates. Thus, slight score increases should not be overinterpreted; they largely reflect the coarse granularity of current evaluation standards.
> > > >
> > > > **3. SkipVAR indeed provides a small but explainable quality benefit by preventing late-step noise on frequency-robust samples.**
> > > > While SkipVAR is designed for acceleration, we consistently observe that, for simple or low-detail images, the final VAR steps may introduce unnecessary high-frequency noise rather than refinements. By skipping these redundant steps, SkipVAR effectively halts decoding at the point of semantic convergence, thereby **avoiding late-step noise amplification**. This mechanism is aligned with our frequency-aware design and explains why slight improvements appear in some metrics, even though the changes are small and within evaluation noise.
> > > >
> > > > **In summary**, the score variations in Tables 2 are dominated by HPSv2’s intrinsic randomness, compounded by the limited sensitivity of existing benchmarks. At the same time, SkipVAR has a real and interpretable ability to reduce late-step noise on frequency-robust samples, which naturally produces the minor metric improvements observed in several categories.

---

### Meta-Review · Area_Chair_omST · 2026-01-08

**Summary:**

The paper proposes to reduce the high inference latency of VAR models via mitigation of computational redundancy. The main concern raised during the initial review process includes the lack of generalization to realistic images, limited use cases, hyperparameter sensitivity like decision setps and threshold. The authors provided additional results and clarity on majority of these. However, the results still remains non convincing as the authors acknowledged the limited utility of SkipVAR-hybrid on sketch type images. Additionally, the reviewers raised concerns on the novelty of the work in relation to existing works on layer skipping like SkipGPT and deeper layer removal. The results from the authors suggest the validity of this argument. Thus, in summary I suggest a borderline reject.

**Reviewer Concerns:**

Limited contribution with respect to existing works on layer skipping, and limited generalization.

**Reviewer Scores:**

Reviewer LJ9B: 4
Reviewer webr: 6
Reviewer HFYL: 6
Reviewer euwc: 4

---

### Decision · Program_Chairs · 2026-01-26

Reject